# Blood-based epigenome-wide analyses of chronic low-grade inflammation across diverse population cohorts

## Graphical abstract

## Authors

Robert F. Hillary, Hong Kiat Ng, Daniel L. McCartney, ..., Riccardo E. Marioni, Paul D. Yousefi, Matthew Suderman

## Correspondence

riccardo.marioni@ed.ac.uk (R.E.M.), paul.yousefi@bristol.ac.uk (P.D.Y.), matthew.suderman@bristol.ac.uk (M.S.)

## In brief

Hillary et al. use DNA methylation data from six population cohorts and state-of-the-art algorithms to develop a predictor of C-reactive protein (CRP) levels. Their predictor performs consistently across diverse populations, outperforms blood CRP and existing algorithms in associations with cardiometabolic health outcomes, and thereby refines opportunities to probe chronic inflammation.

## Highlights

- Existing DNAm predictors explain 10% of variance in circulating CRP levels

- An elastic-net-based predictor outperforms existing models and explains 20% of variance

- Newly described predictor is consistent across life course and ancestries

- This predictor outperforms CRP and existing methods in health outcome associations

Hillary et al., 2024, Cell Genomics 4, 100544
May 8, 2024 © 2024 The Authors. Published by Elsevier Inc.

# Cell Genomics

CellPress

## Article

# Blood-based epigenome-wide analyses of chronic low-grade inflammation across diverse population cohorts

Robert F. Hillary,[1] Hong Kiat Ng,[2] Daniel L. McCartney,[1] Hannah R. Elliott,[3,4] Rosie M. Walker,[1,5] Archie Campbell,[1] Felicia Huang,[6] Kenan Direk,[7] Paul Welsh,[8] Naveed Sattar,[8] Janie Corley,[9] Caroline Hayward,[1,10] Andrew M. McIntosh,[1,11] Cathie Sudlow,[12,13,14] Kathryn L. Evans,[1] Simon R. Cox,[9] John C. Chambers,[2,15] Marie Loh,[2,15,16,17] Caroline L. Relton,[3,4] Riccardo E. Marioni,[1,*] Paul D. Yousefi,[3,4,*] and Matthew Suderman[3,4,18,*]

[1]Centre for Genomic and Experimental Medicine, Institute of Genetics and Cancer, University of Edinburgh, Edinburgh EH4 2XU, UK
[2]Lee Kong Chian School of Medicine, Nanyang Technological University, Clinical Sciences Building, Singapore 308232, Singapore
[3]MRC Integrative Epidemiology Unit at the University of Bristol, Bristol BS8 2BN, UK
[4]Population Health Sciences, Bristol Medical School, University of Bristol, Bristol BS8 1UD, UK
[5]School of Psychology, University of Exeter, Exeter EX4 4QG, UK
[6]MRC Unit for Lifelong Health and Ageing, University College London, London WC1E 7HB, UK
[7]Imperial Clinical Trials Unit, School of Public Health, Imperial College London, London SW7 2AZ, UK
[8]School of Cardiovascular and Metabolic Health, BHF Glasgow Cardiovascular Research Centre, University of Glasgow, Glasgow G12 8TA, UK
[9]Lothian Birth Cohort Studies, Department of Psychology, University of Edinburgh, Edinburgh EH8 9JZ, UK
[10]Medical Research Council Human Genetics Unit, Institute of Genetics and Cancer, University of Edinburgh, Edinburgh EH4 2XU, UK
[11]Division of Psychiatry, University of Edinburgh, Royal Edinburgh Hospital, Edinburgh EH10 5HF, UK
[12]Centre for Clinical Brain Sciences, Edinburgh Imaging and UK Dementia Research Institute, University of Edinburgh, Edinburgh EH16 4SB, UK
[13]British Heart Foundation Data Science Centre, Health Data Research UK, London NW1 2BE, UK
[14]Health Data Research UK, London NW1 2BE, UK
[15]Department of Epidemiology and Biostatistics, School of Public Health, Imperial College London, St Mary's Campus, London W2 1PG, UK
[16]National Skin Centre, Singapore 308205, Singapore
[17]Genome Institute of Singapore, Agency for Science, Technology and Research, Singapore 138672, Singapore
[18]Lead contact
*Correspondence: riccardo.marioni@ed.ac.uk (R.E.M.), paul.yousefi@bristol.ac.uk (P.D.Y.), matthew.suderman@bristol.ac.uk (M.S.)

## SUMMARY

Chronic inflammation is a hallmark of age-related disease states. The effectiveness of inflammatory proteins including C-reactive protein (CRP) in assessing long-term inflammation is hindered by their phasic nature. DNA methylation (DNAm) signatures of CRP may act as more reliable markers of chronic inflammation. We show that inter-individual differences in DNAm capture 50% of the variance in circulating CRP ($N$ = 17,936, Generation Scotland). We develop a series of DNAm predictors of CRP using state-of-the-art algorithms. An elastic-net-regression-based predictor outperformed competing methods and explained 18% of phenotypic variance in the Lothian Birth Cohort of 1936 (LBC1936) cohort, doubling that of existing DNAm predictors. DNAm predictors performed comparably in four additional test cohorts (Avon Longitudinal Study of Parents and Children, Health for Life in Singapore, Southall and Brent Revisited, and LBC1921), including for individuals of diverse genetic ancestry and different age groups. The best-performing predictor surpassed assay-measured CRP and a genetic score in its associations with 26 health outcomes. Our findings forge new avenues for assessing chronic low-grade inflammation in diverse populations.

## INTRODUCTION

Chronic low-grade inflammation is a common feature of many age-related disease states, including heart disease, stroke, and type 2 diabetes.[1,2] C-reactive protein (CRP) is a sensitive marker of systemic inflammation.[3] However, individual measurements can fluctuate substantially following infection or injury.

Therefore, single time point measurements of CRP in clinical settings may provide an incomplete index of an individual's long-term inflammatory status.[4] Identifying inflammatory biomarkers with enhanced temporal stability could improve patient stratification and facilitate robust health outcome testing.

DNA methylation (DNAm) is a reversible epigenetic mechanism in which methyl groups bind to DNA, most commonly within

cytosine-guanine dinucleotides (CpG sites). DNAm is affected by a confluence of genetic and environmental factors and is known to regulate gene expression levels.[5] Two large-scale epigenome-wide association studies (EWASs) have observed >1,000 CpG sites across the genome that associate with blood CRP levels.[6,7] DNAm predictors of CRP levels have been constructed using weighted linear combinations of these CpG sites and explain approximately 10% of inter-individual variation in circulating CRP.[7,8] They show greater longitudinal stability than measured CRP as well as stronger associations with cognitive and cardiometabolic health outcomes,[6,8–10] indicating they are likely to be robust to short-term CRP fluctuations.

Addressing the following points will be critical to further extending our understanding of how DNAm informs the biology and prediction of CRP. There is a need to estimate the expected proportion of variance in CRP captured by genome-wide DNAm probes. Variance component estimates would guide an upper bound for the amount of variance in CRP that can be captured by DNAm predictors, as well as inform the molecular architecture of CRP regulation. Existing predictors are constructed using weights from individual linear regressions. This approach neglects correlations between CpG sites and unknown confounding influences. Bayesian regression methods and several commonly employed feature selection methods, such as penalized regression and principal-component analysis (PCA), may overcome these limitations.[11,12] Furthermore, studies examining the relationship between DNAm and CRP are primarily restricted to adults of European ancestry. It is unclear whether DNAm predictors of CRP are generalizable to other stages of the life course (e.g., childhood and later life) or across genetically diverse individuals.

We address four primary objectives by leveraging blood-based methylation and CRP measurements across six diverse cohorts ($N_{range}$ = 170–17,936). First, we bolster biological insights into the relationship between DNAm and CRP by conducting an EWAS of CRP in the family-based study Generation Scotland (GS; $N$ = 17,936). We also employ two complementary methods, restricted maximum likelihood estimation and Bayesian penalized regression, to estimate the proportion of inter-individual variation in CRP attributable to genome-wide DNAm.[11,13] Second, we address prediction efforts by applying three common feature selection and transformation methods to develop DNAm predictors of CRP levels. They are elastic net regression, Bayesian penalized regression, and PCA. Third, we compare their predictive performances against one another and to existing predictors in the literature. GS serves as the training cohort in prediction analyses. Five diverse test cohorts are employed: Avon Longitudinal Study of Parents and Children (ALSPAC; mother and child pairs), Health for Life in Singapore (HELIOS; adults of self-reported Chinese, Malay, or Indian ethnicity), Southall and Brent Revisited (SABRE; adult males of self-reported European or South Asian ethnicity), and the Lothian Birth Cohorts of 1921 and 1936 (LBC1921 and LBC1936; community-dwelling older adults). Fourth, we compare DNAm predictors of CRP against assay-measured CRP in their associations with 26 cardiometabolic risk factors and health outcomes. Figure 1 shows a visual summary of the study design.

## RESULTS

### Study characteristics and demographics
The six cohorts included in this study had disparate sample sizes and demographic profiles (Table 1). The number of individuals included in analyses ranged from 49 (at age 90 in LBC1921) to 17,936 (age range 18–99 years in GS). Mean CRP levels (mg/L) increased across the life course from 0.2 at birth (cord blood in ALSPAC) to 8.4 at age 90 years (LBC1921).

### EWAS identifies individual CpG sites associated with CRP levels
We first investigated marginal associations between log-transformed blood CRP levels and 752,722 CpG sites in GS ($N$ = 17,936). There were 33,939 associations with $p < 3.6 \times 10^{-8}$ in a basic linear model that adjusted for chronological age, sex, estimated white blood cell (WBC) proportions, and experimental batch (Table S1). The threshold of $p < 3.6 \times 10^{-8}$ represents a commonly employed threshold in EWAS.[14] The lambda or genomic inflation factor for this model was 3.6. Mixed-effects models that included a kinship matrix were used to account for relatedness as sensitivity analyses.[15] Approximately 75% of associations remained associated ($n$ = 25,634 with $p < 3.6 \times 10^{-8}$) after accounting for relatedness, and the remainder of associations had $p < 1 \times 10^{-4}$. Only 2,805 (8.3%) associations had $p < 3.6 \times 10^{-8}$ in a fully adjusted model that further accounted for five lifestyle factors and population structure (Figure 2A; Table S2). The lambda value for this fully adjusted model was 1.7. Q-Q plots for the basic and fully adjusted models are shown in Figure S1. Sensitivity analyses showed that body mass index (BMI) alone attenuated 72.5% of associations from the basic model to non-significance. Effect sizes for associations from the basic model were attenuated, on average, by 40%. Smoking behavior[16] attenuated 51.5% of associations to non-significance, and effect sizes were attenuated, on average, by 20.2%. Covarying for both BMI and smoking behavior (and not other lifestyle factors) attenuated 84.9% of associations from the basic model to non-significance. The substantial attenuation observed after accounting for lifestyle behaviors, and in particular BMI and smoking, highlights their strong impact on DNAm and chronic low-grade inflammation.[6] The confounding influences of the observed lifestyle factors and further undocumented confounders also likely underscore the inflation observed in our linear regression models.

A look-up analysis using the EWAS Catalog revealed that 1,496 associations (4.4%) from the basic model were previously reported in the literature.[6,7,17–19] Effect sizes for 1,379 significant CpG associations in a recent EWAS on CRP were correlated 97% with corresponding associations in our study (Figure S2).[6] Lastly, we repeated the basic model using Bayesian penalized regression, which can better account for correlations among probes and control for inflation.[11] This method identified 47 lead CpG sites with a posterior inclusion probability greater than 80%, 38 of which had $p < 3.6 \times 10^{-8}$ in the linear model (STAR Methods; Table S3). Our EWAS findings show strong agreement with the existing literature and highlight that a small subset of densely correlated regions show robust associations with CRP levels.

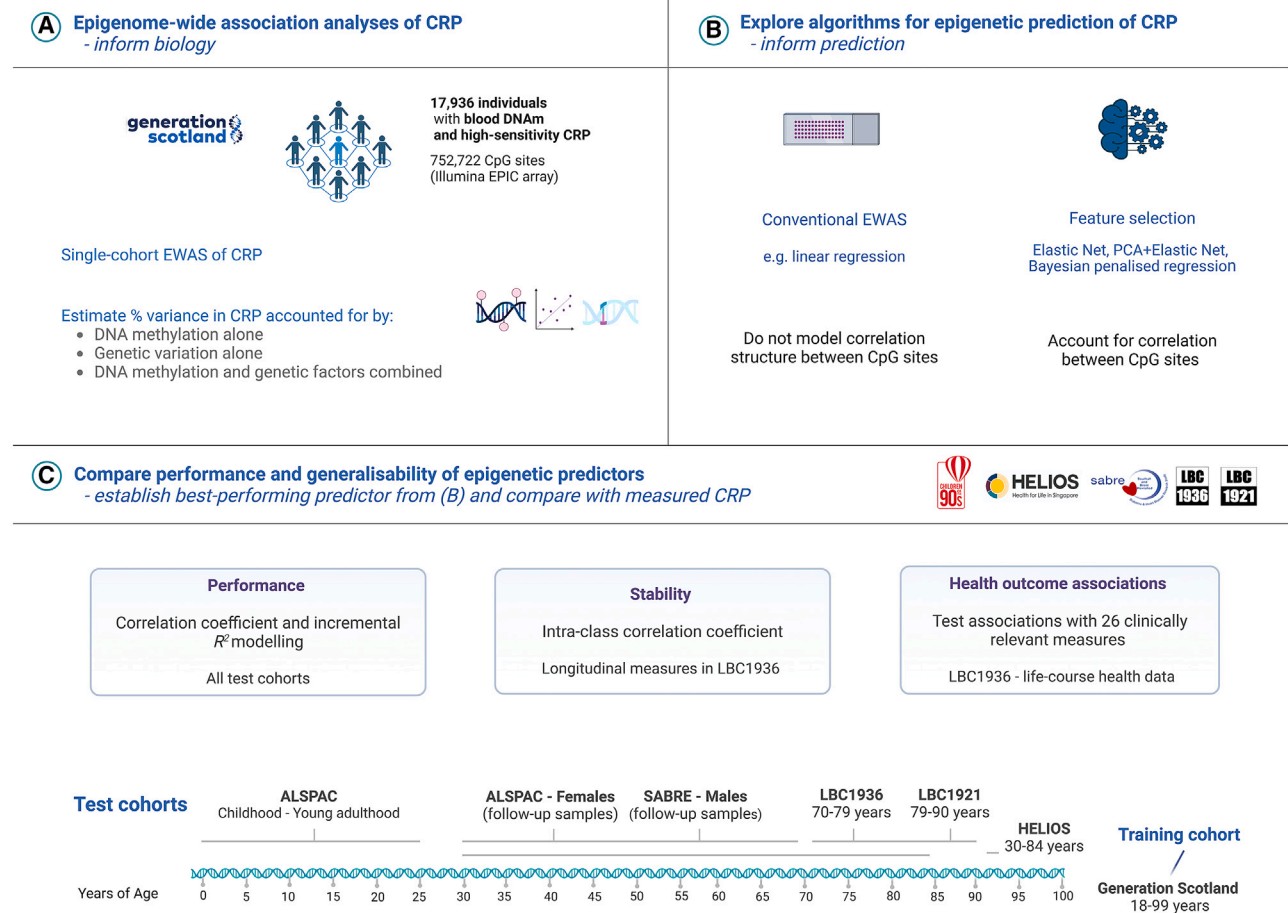

**Figure 1. Blood epigenome-wide analyses of CRP levels across a diverse set of population cohorts**

(A) There were 17,936 individuals in Generation Scotland with complete high-sensitivity CRP measurements and genome-wide DNAm profiling. This allowed for an epigenome-wide scan of associations between differential DNAm and blood CRP levels, alongside a variance component analysis of molecular phenotypes and CRP.

(B) A suite of feature selection and transformation methods were implemented to develop new DNAm predictors of CRP. These methods account for the correlation structure between features (CpG sites) and may offer improved predictive performances over existing methods (i.e., methylation risk scores with weights from linear regression models).

(C) The predictive performances of CRP predictors derived from feature selection methods in (B) were compared against existing predictors. The five test cohorts harbored cross-sectional samples that encompass the life course (i.e., cord blood samples and childhood through to later life), adult males and females, and individuals from different ethnic backgrounds and countries of residency. Of the test cohorts, the Lothian Birth Cohort 1936 was selected for health outcome testing, given that the study population was at elevated risk for age-related disease states when compared to other cohorts and subgroups. It also constituted a larger analytical sample than the Lothian Birth Cohort 1921. ALSPAC, Avon Longitudinal Study of Parents and Children; CpG, cytosine-phosphate-guanine dinucleotide; CRP, C-reactive protein; DNAm, DNA methylation; EWAS, epigenome-wide association study; GS, Generation Scotland; HELIOS, Health for Life in Singapore; LBC1921, Lothian Birth Cohort 1921; LBC1936, Lothian Birth Cohort 1936; SABRE, Southall And Brent Revisited. Image was created with BioRender.com.

## Proportion of variance in CRP levels captured by global DNAm and genetic factors

Next, we assessed whether global patterns of DNAm were associated with individual differences in blood CRP profiles within GS ($N$ = 17,936). BayesR+ was used to perform Bayesian penalized regression and Gaussian mixture-based variance partitioning.[11] Variance estimates can be affected by the covariates used in the associated models. Log-transformed CRP levels were adjusted for age and sex prior to entry in variance partitioning analyses. CpG β values were also adjusted for age, sex, WBC proportions,

and experimental batch. The proportion of variance in CRP levels captured by genome-wide DNAm alone was 51.7% (95% credible interval [CrI]: 48.0%, 55.0%), guiding an upper bound for the performance of DNAm CRP predictors.

Unlike genetic (or single-nucleotide polymorphism [SNP]-based) heritability estimates, the variance captured by DNAm probes may reflect both cause and consequence on the phenotype. Disentangling the independent and combined contributions from genetic and DNAm variation would refine insights into the molecular architecture of CRP. Using BayesR+, the

**Table 1. Summary of demographics and C-reactive protein measurement across six diverse cohorts**

| Sample | N | Age (years), mean, SD | Sex, N female, % female | CRP (mg/L), mean, SD | CRP assay | DNAm assay (Illumina) |
|---|---|---|---|---|---|---|
| ALSPAC | | | | | | |
| Age 0 | 389 | – | 198, 50.1 | 0.2, 0.9 | high sensitivity | 450 K array |
| Age 9 | 336 | 9.8, 0.3 | 163, 48.5 | 0.6, 1.1 | high sensitivity | 450 K array |
| Age 15 or 17 | 945 | 17.1, 1.1 | 492, 52.1 | 1.1, 1.8 | high sensitivity | 450 K array |
| Age 24 | 745 | 24.5, 0.8 | 365, 49.0 | 1.7, 2.5 | high sensitivity | 450 K array |
| Mothers (follow-up) | 773 | 47.8, 4.3 | 773, 100 | 1.8, 2.2 | high sensitivity | 450 K array |
| Generation Scotland | | | | | | |
| Whole sample | 17,936 | 47.5, 14.9 | 10,536, 58.7 | 3.3, 2.9 | high sensitivity | EPIC array |
| HELIOS | | | | | | |
| Chinese ethnicity | 1,778 | 54.6, 11.8 | 729, 61.9 | 1.5, 3.8 | wide range | EPIC array |
| Malay ethnicity | 242 | 55.4, 11.1 | 148, 61.2 | 2.9, 4.9 | wide range | EPIC array |
| Indian ethnicity | 225 | 51.3, 11.8 | 126, 56.0 | 4.4, 7.6 | wide range | EPIC array |
| SABRE | | | | | | |
| European ethnicity | 315 | 69.9, 6.3 | 0, 0 | 3.6, 7.9 | high sensitivity | 450 K array |
| South Asian ethnicity | 273 | 69.3, 6.2 | 0, 0 | 2.7, 3.8 | high sensitivity | 450 K array |
| LBC1936 | | | | | | |
| Age 70 (W1) | 885 | 69.6, 0.8 | 439, 49.6 | 5.3, 6.9 | low sensitivity | 450 K array |
| Age 73 (W2) | 756 | 72.5, 0.7 | 361, 47.8 | 3.0, 5.5 | high sensitivity | 450 K array |
| Age 76 (W3) | 536 | 76.3, 0.7 | 258, 48.1 | 3.1, 4.8 | high sensitivity | 450 K array |
| Age 79 (W4) | 492 | 79.3, 0.6 | 240, 48.8 | 2.5, 5.6 | high sensitivity | 450 K array |
| LBC1921 | | | | | | |
| Age 87 (W3) | 170 | 86.6, 0.4 | 92, 54.1 | 5.5, 7.2 | low sensitivity | 450 K array |
| Age 90 (W4) | 49 | 90.2, 0.1 | 29, 59.2 | 8.4, 11.7 | low sensitivity | 450 K array |

Low-sensitivity assays could not reliably detect values below 3 mg/L. These values were set to 1.5 mg/L in line with previous LBC publications[8] and may have contributed to altered distribution properties at time points and cohorts that used low-sensitivity platforms. ALSPAC, Avon Longitudinal Study of Parents and Children; CRP, C-reactive protein; DNAm, DNA methylation; HELIOS, Health for Life in Singapore; LBC, Lothian Birth Cohort; SABRE, Southall and Brent Revisited; SD, standard deviation; W, wave.

proportion of variance explained by genetic variation alone was 13.4% (95% CrI: 11.6%, 15.5%), aligning well with a recent SNP-based heritability estimate of 13% from an existing GWAS on CRP.[20]

Using BayesR+, the joint variance captured by genetics and DNAm was 61.0% (95% CrI: 57.6%, 64.0%). The contribution of DNAm to this estimate was 49.0%, which is similar to the estimate from DNAm analysis alone (51.7%) and suggests it was largely independent from underlying genetic factors. Sensitivity analyses were performed in OSCA using a linear mixed-model approach with an epigenetics relationship matrix.[13] The same model structures were applied to the BayesR+ and OSCA strategies. Estimates from OSCA were highly consistent with those from the Bayesian strategy, as illustrated in Figure 2B (Table S4).

**Comparing feature selection methods in developing DNAm predictors of CRP**

We focused on five distinct methods to generate DNAm predictors of CRP. We trained three predictors using (1) elastic net regression, (2) Bayesian penalized regression, and (3) PCA with elastic net regression (PCA+elnet) (3) (N = 17,936). We also derived two additional predictors using EWAS weights from (4)

Wielscher et al.[6] and (5) our own linear EWAS. We then projected them into ALSPAC, HELIOS, SABRE, LBC1936, and LBC1921 (summary data are presented in Table S5).

We note that the training cohort and test cohorts showed variation in terms of the arrays used to measure CRP levels and blood DNAm profiles (Table 1). In the EWAS stage, we utilized all CpGs available in GS in order to document as completely as possible the relationships between local and global DNAm variation and CRP blood levels. In the prediction stage, we aimed to ensure maximum generalizability to other and older cohorts. Unless otherwise stated, we first restricted the training space to CpG sites that were common to the EPIC and 450 K arrays. Specifically, we restricted CpGs to those present in ALSPAC, GS, SABRE, and the LBC cohorts following quality control (n = 374,785 sites; STAR Methods). We also acknowledge that other cohorts may have a slightly different set of CpGs from this list. We held HELIOS as a further external test cohort to understand how missing CpG sites would impact prediction. HELIOS did not contain 193 of these 374,785 sites.

We first assessed the longitudinal stabilities of assay-measured CRP and DNAm CRP (all five predictors) in the LBC1936 (STAR Methods). Longitudinal analyses were restricted to waves 2, 3, and 4, as high-sensitivity CRP

**A**

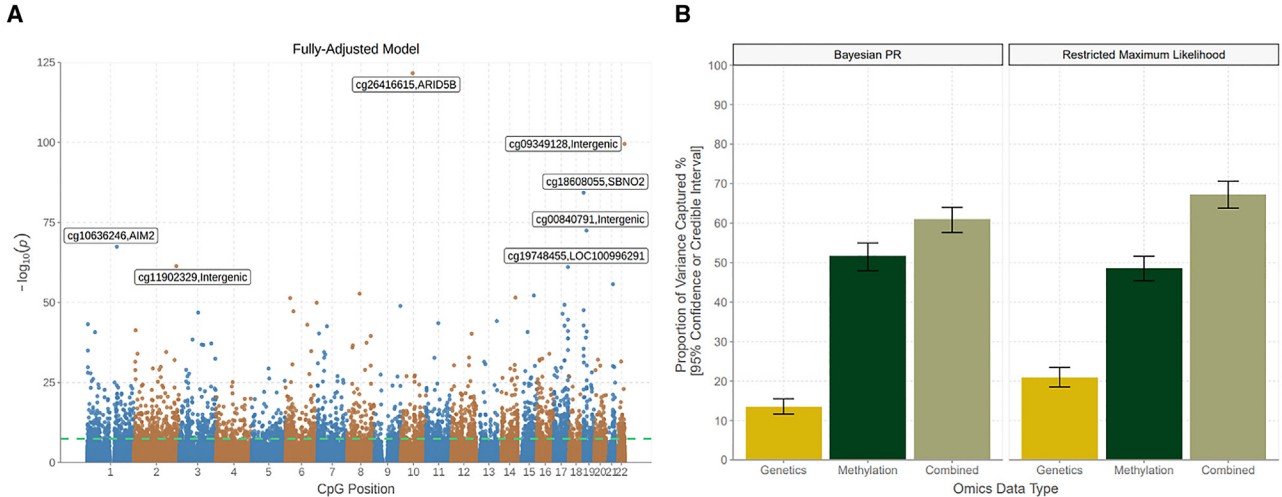

**B**

**Figure 2. Epigenome-wide association and variance component analyses of blood CRP levels in Generation Scotland**

(A) A Manhattan plot shows associations between genome-wide CpG probes and log-transformed CRP levels ($N = 17,936$). Associations from a fully adjusted linear regression model are displayed. The green line denotes the epigenome-wide significance threshold at $p < 3.6 \times 10^{-8}$. The seven strongest associations (smallest $p$ values) are annotated for clarity.

(B) The proportion of variance captured by genome-wide genetic and methylation factors, separately, are shown in gold and dark green bars, respectively. The beige bar details the joint variance captured by genetic and methylation variation when conditioned on one another. Vertical bars denote the 95% credible (Bayesian PR) and confidence (restricted maximum likelihood) intervals, respectively. CpG, cytosine-phosphate-guanine dinucleotide; CRP; C-reactive protein; PR, penalized regression.

measures were only present at these time points, enabling consistent comparisons. DNAm predictors showed higher intra-class correlation coefficients (0.84–0.94) than assay-measured CRP (0.78), indicating greater stabilities (Table S6). Linear mixed models were then used to assess whether assay-measured and DNAm CRP derived at wave 2 showed longitudinal associations with CRP levels over subsequent waves (i.e., in effect, predicting CRP profiles). Assay-measured CRP showed the strongest association with repeat measurements (interaction term between wave 2 CRP and age: $\beta = -0.20$, $p < 1.6 \times 10^{-38}$) (Table S7; STAR Methods). The elastic net-, Bayesian-, and PCA+elnet-based predictors also exhibited significant, albeit weaker, associations ($\beta = -0.05$ for all, range of $p = [3.9 \times 10^{-3}, 7.7 \times 10^{-3}]$). Neither EWAS-based predictor showed a strong association (Table S7). The longitudinal decline in CRP levels in the LBC1936 is apparent in Table 1 and may reflect attrition bias.

Next, we evaluated correlations between DNAm CRP and assay-measured CRP in all cohorts to assess the best-performing prediction method. Predictors built using elastic net regression, Bayesian penalized regression, and PCA+elnet showed comparable Pearson's correlation coefficients to one another in adult samples. Their correlation coefficients ranged from 0.27 to 0.53, 0.28 to 0.49, and 0.28 to 0.47 across cohorts, respectively (Figure 3A; Table S8). Correlations were consistent between males and females and, in HELIOS and SABRE, across genetically diverse individuals. Notably, correlations were weaker in childhood samples from ALSPAC including at ages 0 ($r = \sim0.07$) and 9 years ($r = \sim0.20$). These analyses suggested that DNAm predictors of CRP were robust to differences in sex and ethnicity but less so to age variation. HELIOS showed similar

patterns to other cohorts, which suggested that missingness of CpG sites did not impact performance.

The linear predictor—analogous to a polygenic score—based on association effect sizes in the Wielscher EWAS[6] ($p < 1 \times 10^{-7}$, $n \leq 1,379$ sites; STAR Methods) tended to be less correlated with assay-measured CRP ($r = \sim0.2$). Retraining the predictor using PCA+elnet (and focusing on the same CpG sites) almost doubled the correlations (Figure 3A). There was a negligible correlation between CRP levels and a weighted score derived from our EWAS ($n \leq 33,939$ sites with $p < 3.6 \times 10^{-8}$), likely reflecting random noise from many more weakly associated individual sites.

The same patterns held true for incremental $R^2$ estimates of CRP variance explained beyond age, sex, and genotype. A predictor from elastic net regression explained 17.7% of the variance in CRP over age and sex, with comparable estimates from PCA+elnet and Bayesian approaches (Figure 3B). EWAS-based predictors explained only 0%–4% of phenotypic variance. A genetic score explained 5.1% of the variance in CRP. DNAm predictors captured variance in CRP independently from the genetic predictor, consistent with the variance partitioning analyses.

**Elastic net regression is sufficient to enhance DNAm prediction of CRP**

The primary PCA+elnet-based approach was applied to 1,379 sites that associated with CRP at $p < 1.0 \times 10^{-7}$ in the Wielscher et al. study.[6] The strategy was repeated across a range of pre-filtering $p$ value thresholds to determine its sensitivity to specific thresholds ($p < 0.05$ as the least stringent to $p < 1 \times 10^{-20}$ as the most stringent). The PCA+elnet predictor performed similarly

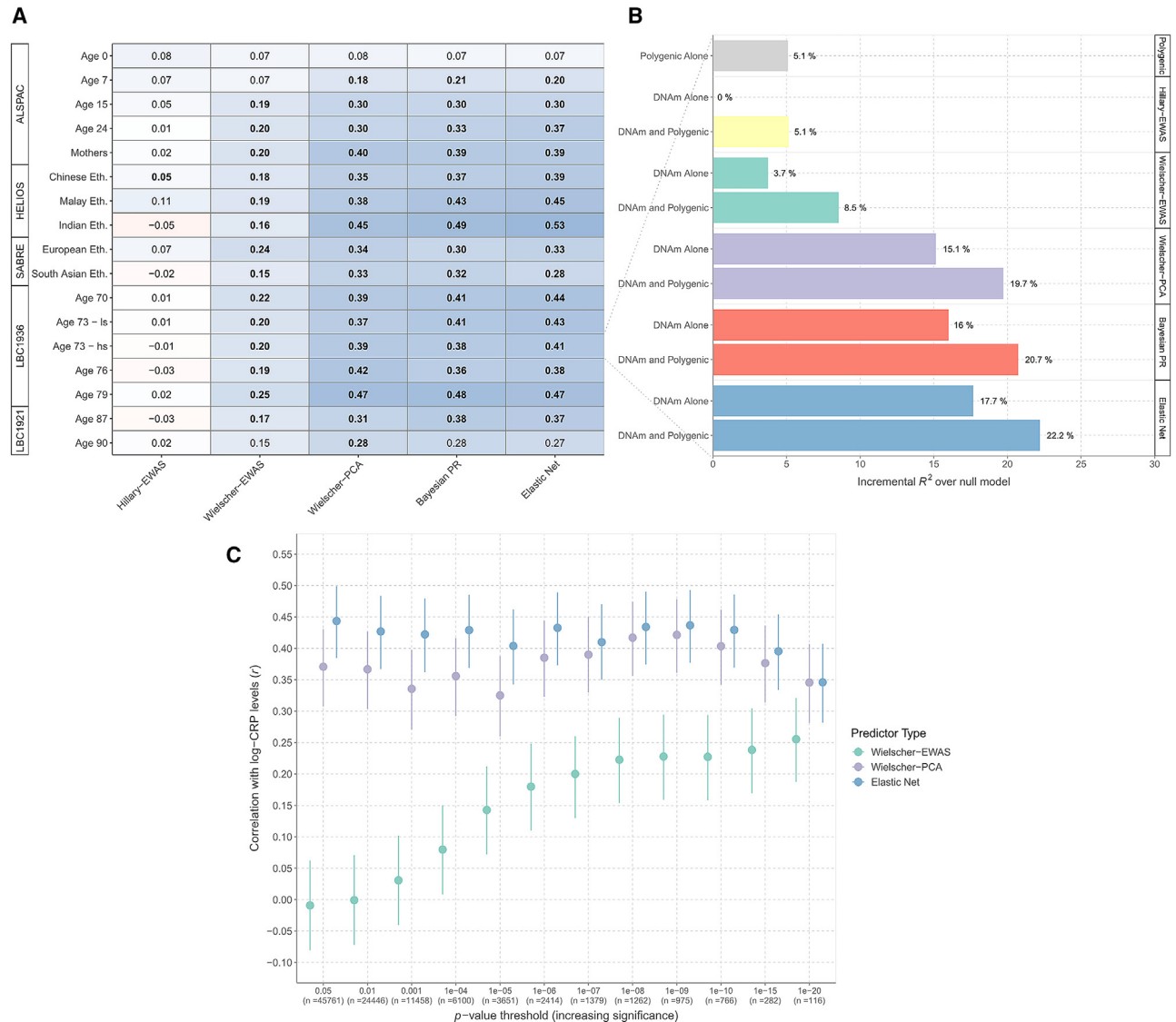

**Figure 3. DNAm prediction of blood CRP levels using five separate strategies**

(A) Pearson's correlation coefficients between log-transformed CRP levels and five different DNAm predictors of circulating levels. Weighted linear DNAm predictors for CRP levels were derived from (1) elastic net regression (Elastic Net), (2) Bayesian penalized regression (Bayesian PR), (3) PCA combined with elastic net regression (Wielscher-PCA[6]), (4) an EWAS by Wielscher et al. (Wielscher-EWAS[6]), and (5) the present EWAS (Hillary-EWAS). Low-sensitivity and high-sensitivity CRP measures were available at age 73 (wave 2) of the LBC1936 and are included in this plot to enable cross-assay comparison. The high-sensitivity measures alone are reported in the main text for this time point.

(B) The proportion of variance captured in log-transformed CRP levels by a polygenic score alone and DNAm CRP from (A) are shown for wave 2 of the LBC1936 (N = 756, incremental $R^2$ estimates above null model, see main text). An additive genetic and DNAm model is also shown for each of the five prediction strategies.

(C) The PCA and elastic net regression method in the main text relied on pre-filtering sites to those that surpassed genome-wide significance in the Wielscher et al. EWAS[6] (i.e., $p < 1.0 \times 10^{-7}$). The method was then repeated using different $p$ value thresholds to filter probes prior to PCA. The resulting predictors were compared against (1) weighted linear combinations using EWAS weights alone and (2) elastic net regression on the filtered CpGs (i.e., bypassing the PCA step). Pearson's correlations were computed between log-transformed CRP and DNAm CRP for all three methods and for $p$ value thresholds with increasing stringency. Vertical lines denote the 95% confidence interval. ALSPAC, Avon Longitudinal Study of Parents and Children; CpG, cytosine-phosphate-guanine dinucleotide; CRP, C-reactive protein; DNAm, DNA methylation; Eth., ethnicity; EWAS, epigenome-wide association study; HELIOS, Health for Life in Singapore; LBC1936, Lothian Birth Cohort 1936; PCA, principal-component analysis; PR, penalized regression.

regardless of the threshold ($r$ = 0.33–0.42). We also trained the predictor using sites common to our cohorts ($n$ = 374,785) to directly match the original strategy used by Higgins Chen et al.[12] The resulting correlation coefficient was slightly lower ($r$ = 0.30).

By contrast, the performance of the EWAS-based predictor tended to improve at more stringent $p$ value thresholds. The correlation coefficient ranged from −0.01 (using sites at $p < 0.05$) to 0.26 ($p < 1 \times 10^{-20}$) (Figures 3C; Table S9).

Additionally, we repeated the approach without the PCA step and retrained predictors using elastic net regression alone. Elastic net alone outperformed the combined PCA+elnet strategy, on average, by 10%, suggesting that the PCA step did not enhance predictive performance.

## DNAm predictors outperform assay-measured CRP in associations with cardiometabolic health outcomes

Lastly, we compared DNAm CRP and assay-measured CRP in their associations with 26 health outcomes at age 73 years in the LBC1936 (STAR Methods). Previous studies have focused on EWAS-based predictors alone. Here, we focused on the elastic net-based predictor given that it was the best-performing method in correlation analyses and incremental $R^2$ modeling. Overall, assay-measured CRP and DNAm CRP displayed similar relationships with continuous outcomes with 15 and 13 associations (from 21 outcomes) having false discovery rate $p$ ($p_{FDR} < 0.05$, respectively (Figure 4; Table S10).

However, for cardiometabolic disease outcomes, DNAm CRP outperformed assay-measured CRP (Figure 4). Whereas DNAm CRP was strongly associated with history of cardiovascular disease (odds ratio [OR] = 1.28, $p_{FDR} = 6.6 \times 10^{-3}$), hypertension (OR = 1.30, $p_{FDR} = 2.3 \times 10^{-3}$), stroke (OR = 1.63, $p_{FDR} = 3.4 \times 10^{-3}$), and type 2 diabetes (OR = 1.72, $p_{FDR} = 8.0 \times 10^{-5}$), assay-measured CRP was weakly associated only with hypertension (OR = 1.19, $p_{FDR} = 0.03$) and stroke (OR = 1.41, $p_{FDR} = 0.03$). Both measures were similarly associated with risk of all-cause mortality (hazard ratios = 1.47 and 1.45, $p_{FDR} = 3.0 \times 10^{-6}$ and $3.7 \times 10^{-6}$, respectively; Figure 4).

Association patterns were similar for the Bayesian- and PCA+elnet-based predictors, which associated with 19 and 20 outcomes at $p_{FDR} < 0.05$, respectively. The EWAS-based predictors associated with 8 (our EWAS) and 15 (Wielscher EWAS[6]) outcomes (Figures S3–S8). There were no associations for the genetic score following multiple testing correction. Therefore, our DNAm models capture adverse health outcomes better than assay-measured CRP, as well as existing DNAm and genetic models of CRP.

## DISCUSSION

We developed a DNAm predictor of CRP that explains up to 20% of the variance in circulating concentrations, almost doubling that captured by existing predictors.[7,8] DNAm CRP also outperforms genetic scores and assay-measured CRP in association analyses with cardiometabolic disease outcomes and risk factors. While the DNAm predictor was developed using data from over 17,000 Scottish adults, it is generalizable across cohorts with distinct birth periods, DNAm normalization pipelines, sex differences, and individuals of different ethnic backgrounds. These analyses comprehensively outline the utility of DNAm as a biomarker of chronic low-grade inflammation and provide new opportunities to capture inflammatory burden across diverse populations.

We identified elastic net regression as the best-performing method to proxy chronic low-grade inflammation from DNAm. Recently, Higgins Chen et al. showed that PCA prior to elastic net regression improved the reliability of epigenetic age estima-

tors.[12] Similarly, Doherty et al. found that PCA in advance of elastic net regression outperformed 12 other strategies, including elastic net regression alone, in the DNAm prediction of telomere length.[21] Trejo-Banos et al. showed that DNAm predictors of BMI and cigarette smoking developed using Bayesian penalized regression captured more phenotypic variance than those from conventional penalized regression methods.[11,22] Here, elastic net regression without PCA offered a slight increase in performance over these methods (~1%–2%) in explaining CRP variance. We show that this benefit holds over a range of pre-filtering criteria. The method also offers greater interpretability by having fewer selected features and lower computational expense than the PCA-based approach. Furthermore, it provides faster run times than BayesR+ (minutes versus hours based on current software versions). Nevertheless, the optimal method for a given trait is likely to depend on the precise relationship between the phenotype and molecular dataset in question.

Existing DNAm predictors of inflammation represent additive weighted scores that consider CpG sites and weights (i.e., coefficients) from EWASs alone. This approach is analogous to the development of polygenic scores. Verschoor et al.[10] showed that an aggregate score derived from the recent EWAS by Wielscher et al.[6] outperformed assay-measured CRP as a marker of cardiopulmonary disease and long-term health status. However, neither measure associated with all-cause mortality, which is in contrast to our study.[10] A seven-CpG score derived from an earlier EWAS by Ligthart et al. has also been associated with a wide range of neurocognitive health outcomes in adults and neonates.[7–9,23–25] These aggregate EWAS scores have shown promise in correlating with cardiometabolic disease risk and risk factors.[6,7] However, we show that predictors or measures from feature selection methods (i.e., elastic net regression) capture much more trait variance and associate with a greater number of outcomes. Our predictors performed comparably in residents of the UK and Singapore with diverse self-report ethnicities, which indicates its potential as a biomarker of inflammation in different populations. In contrast to polygenic scores, research into the portability of DNAm-based predictors is limited. There remains a need to expand prediction efforts to individuals across other global regions and of additional ethnic backgrounds and ancestries in order to capture a fuller range of genetic and environmental contexts. It is also essential to include target populations with disparate genetic distances in the training sample in order to reliably document the utility of DNAm-based prediction. Our predictors were developed using whole-blood adult samples but performed poorly at the extremes of the life course, including in neonates and in the ninth decade of life. Indeed, the predictors had near-zero correlations in cord blood samples. Cord blood DNAm serves a better proxy for cord blood CRP than maternal blood DNAm.[26] Therefore, additional tissue sources may be required to improve the generalizability of inflammatory biomarkers alongside environmental, technical, and statistical considerations.

Genome-wide DNAm serves as a strong proxy for chronic low-grade inflammation, capturing up to 50% of inter-individual variation in circulating CRP levels. Our DNAm predictors of CRP are generalizable across the population cohorts tested and offer new and improved opportunities to examine the association between

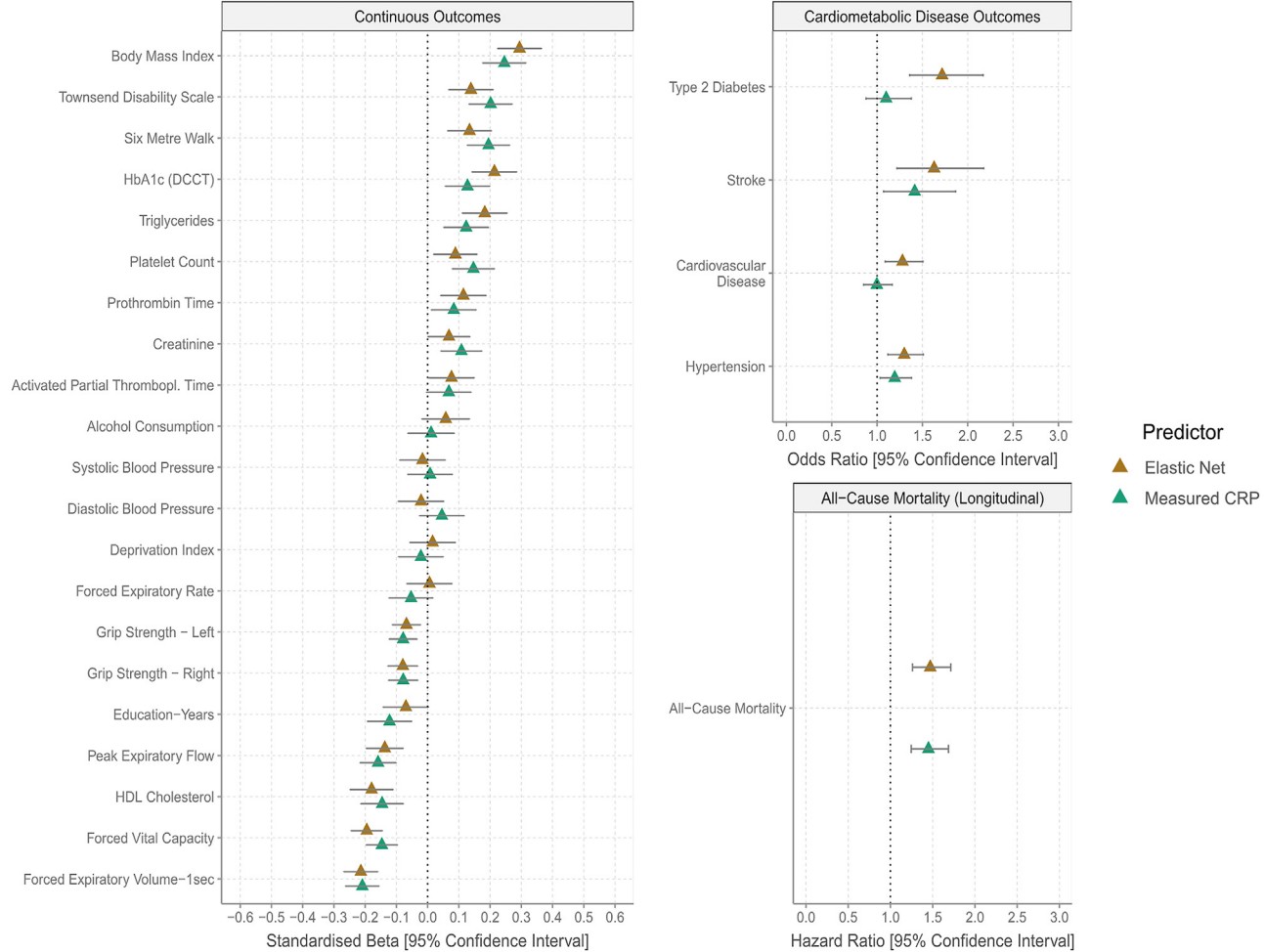

**Figure 4. Associations of health outcomes with DNAm CRP from elastic net regression and assay-measured CRP**

Linear and logistic regression models (two-sided) were used to test for cross-sectional associations of DNAm and assay-measured (i.e., phenotypic) CRP with cardiometabolic, lifestyle, and self-report disease variables at wave 2 of the LBC1936 ($N \leq 756$). Cox proportional hazard models tested for associations between CRP (assay measured or DNAm) derived at wave 2 and time to death due to all-cause mortality. Here, only DNAm CRP from elastic net regression was utilized, given that it was deemed the best-performing method in correlation analysis and incremental $R^2$ modeling. Association tests using DNAm CRP from other prediction strategies and a polygenic score for CRP are shown in Figures S3–S8. CRP, C-reactive protein; DCCT, Diabetes Control and Complications Trial; DNAm, DNA methylation; HDL, high-density lipoprotein; Thrombopl., thromboplastin.

chronic inflammation and health outcomes across disparate clinical and regional populations.

### Limitations of the study

Limitations of this study include the non-consideration of medication data and cross-sectional measurements preventing evaluation of the capacity of CRP models to predict health outcomes. The non-consideration of medication data and the advanced age of the LBC1936 cohort could have influenced associations between measured CRP and cross-sectional health outcomes.

Additional limitations include different platforms for methylation typing and CRP measurements, small numbers of analytical samples in some cohorts (e.g., LBC1921), and the potential for non-linear associations between CRP and CpG methylation, which we did not address. Future work should focus on consid-

ering further advanced statistical methodologies that can consider nuanced and complex relationships between DNAm and CRP in addition to other human traits.

### STAR★METHODS

Detailed methods are provided in the online version of this paper and include the following:

- KEY RESOURCES TABLE
- RESOURCE AVAILABILITY
  - Lead contact
  - Materials availability
  - Data and code availability
- EXPERIMENTAL MODEL AND STUDY PARTICIPANT DETAILS
  - Training cohort: Generationy Scotland
  - Test cohorts
  - Avon longitudinal study of children and parents

- ○ Health for life in Singapore
- ○ Southall And Brent REvisited
- ○ The Lothian Birth Cohorts of 1921 and 1936
- ● METHOD DETAILS
  - ○ Epigenome-wide association studies on CRP levels
  - ○ Bayesian penalized regression
  - ○ Estimating the proportion of variance in CRP levels attributable to genetics and DNAm
  - ○ DNAm prediction of CRP levels
  - ○ Evaluation of DNAm predictors of CRP
  - ○ Assessing longitudinal stability of CRP measures
  - ○ Assessing relationship between assay-measured or DNAm CRP and future CRP measurements
  - ○ Health outcome association tests
  - ○ Polygenic scoring in the LBC1936
- ● QUANTIFICATION AND STATISTICAL ANALYSIS

## SUPPLEMENTAL INFORMATION

## ACKNOWLEDGMENTS

This research was funded in whole, or in part, by Wellcome (104036/Z/14/Z, 220857/Z/20/Z, 217065/Z/19/Z, 067100, 37055891, 086676/7/08/Z, and 221890/Z/20/Z). For the purpose of open access, the author has applied a CC BY public copyright licence to any author-accepted manuscript version arising from this submission. We are extremely grateful to all participants, team members, and support staff in GS, HELIOS, SABRE, and the LBCs for their invaluable contributions to this study. Furthermore, for ALSPAC, we are extremely grateful to all the families who took part in this study, the midwives for their help in recruiting them, and the whole ALSPAC team, which includes interviewers, computer and laboratory technicians, clerical workers, research scientists, volunteers, managers, receptionists, and nurses. GS received core support from the Chief Scientist Office of the Scottish Government Health Directorates (CZD/16/6) and the Scottish Funding Council (HR03006). Genotyping and DNAm profiling of the GS samples was carried out by the Genetics Core Laboratory at the Edinburgh Clinical Research Facility, Edinburgh, Scotland, and was funded by the Medical Research Council UK and Wellcome (Wellcome Trust Strategic Award Stratifying Resilience and Depression Longitudinally [STRADL; reference 104036/Z/14/Z]). The DNAm data assayed for GS were partially funded by Wellcome (220857/Z/20/Z), a 2018 NARSAD Young Investigator Grant from the Brain & Behavior Research Foundation (27404; awardee: Dr. David M. Howard), and a JMAS SIM fellowship from the Royal College of Physicians of Edinburgh (awardee: Dr. Heather C. Whalley). Roche Diagnostics supported this study through the provision of free reagents and a grant for the measurement of CRP in GS. We thank Elaine Butler, Ross Hepburn, and Ellen Macdonald, all of the University of Glasgow, for excellent technical support. The UK Medical Research Council and Wellcome (217065/Z/19/Z) and the University of Bristol provided core support for ALSPAC. A comprehensive list of grant funding is available on the ALSPAC website (http://www.bristol.ac.uk/alspac/external/documents/grant-acknowledgements.pdf). Methylation data in the ALSPAC cohort were generated as part of the UK BBSRC-funded (BB/I025751/1 and BB/I025263/1) Accessible Resource for Integrated Epigenomic Studies (ARIES, https://www.ariesepigenomics.org.uk). The ALSPAC study was further supported by the National Institute for Health and Care Research Bristol Biomedical Research Centre. The views expressed are those of the author(s) and not necessarily those of the NIHR or the Department of Health and Social Care. The HELIOS study was supported by the Singapore Ministry of Health's National Medical Research Council under its OF-LCG funding scheme (MOH-000271-00) and intramural funding from Nanyang Technological University, Lee Kong Chian School of Medicine, and the National Healthcare Group of Singapore. SABRE was supported at baseline by the Medical Research Council, the British Heart Foundation, and Diabetes UK. At follow-up, the SABRE study was funded by Wellcome (067100, 37055891, and 086676/7/08/Z), the British Heart Foundation (PG/06/145, PG/08/103/26133, PG/12/ 29/29497, and CS/13/1/30327), and Diabetes UK (13/0004774). The SABRE study team also acknowledges the support of the National Institute of Health Research Clinical Research Network (NIHRCRN). The LBC1936 is jointly core-funded by the Biotechnology and Biological Sciences Research Council and the Economic and Social Research Council (BB/W008793/1) and received support from Age UK (Disconnected Mind programme), the Milton Damerel Trust, the Medical Research Council (MR/M01311/1), and the University of Edinburgh. LBC1921 data collection was supported by grants from the Biotechnology and Biological Sciences Research Council (15/SAG09977) and the Chief Scientist Office of the Scottish Executive Health Department (CZB/4/505, ETM/55, CZH/4/213, and CZG/3/2/79). Methylation typing was supported by the Centre for Cognitive Ageing and Cognitive Epidemiology (Pilot Fund award), Age UK, The Wellcome Trust Institutional Strategic Support Fund, The University of Edinburgh, and The University of Queensland. R.F.H. is supported by a British Heart Foundation Immediate Fellowship (FS/IPBSRF/22/27042). H.R.E. is supported by the Medical Research Council Integrative Epidemiology Unit at the University of Bristol (MC_UU_00011/5). F.H. and K.D. were supported within a unit that received support from the UK Medical Research Council (MC_UU_12019/1). S.R.C. was supported by a Sir Henry Dale Fellowship jointly funded by Wellcome and the Royal Society (221890/Z/20/Z). R.E.M. is supported by an Alzheimer's Society major project grant (AS-PG-19b-010). P.D.Y. and M.S. are supported by the Medical Research Council Integrative Epidemiology Unit at the University of Bristol (MC_UU_00011/5) and Cancer Research UK (C18281/A29019).

## AUTHOR CONTRIBUTIONS

R.F.H., C.L.R., R.E.M., P.D.Y., and M.S. conceptualized the study design. R.F.H., H.K.N., P.D.Y., and M.S. performed the analyses. D.L.M., H.R.E., R.M.W., A.C., F.H., D.K., P.W., N.S., J.C., C.H., C.S., A.M.M., K.L.E, S.R.C., J.C.C., and M.L. were involved in data generation and preparation. All authors reviewed and approved of the manuscript.

## DECLARATION OF INTERESTS

R.F.H. and R.E.M. act as scientific consultants for Optima Partners. R.E.M. is an advisor to the Epigenetic Clock Development Foundation. R.F.H. has received consultant fees from Illumina. P.W. reports grant income from Roche Diagnostics in relation to and outside of the submitted work, as well as grant income from AstraZeneca, Boehringer Ingelheim, and Novartis outside the submitted work and speaker fees from Novo Nordisk and Raisio outside the submitted work. N.S. has consulted for Afimmune, Amgen, AstraZeneca, Boehringer Ingelheim, Eli Lilly, Hanmi Pharmaceuticals, Merck Sharp & Dohme, Novartis, Novo Nordisk, Pfizer, and Sanofi and has received grant support paid to his university from AstraZeneca, Boehringer Ingelheim, Novartis, and Roche Diagnostics outside the submitted work.

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

**Cell Genomics**
Article

## STAR★METHODS

### KEY RESOURCES TABLE

| REAGENT or RESOURCE | SOURCE | IDENTIFIER |
|---|---|---|
| **Biological samples** | | |
| Human blood DNA | Anonymous Donors | N/A |
| **Critical commercial assays** | | |
| Infinium HumanMethylation450 BeadChip | Illumina | Cat# WG-314 |
| Infinium MethylationEPIC BeadChip | Illumina | Cat# WG-317 |
| Illumina HumanOmniExpressExome-8 | Illumina | Cat #20024677 |
| c311 stand alone system | Roche | Cat# 04826876001 |
| **Deposited data** | | |
| Code used in this manuscript | This manuscript | https://doi.org/10.5281/zenodo.10154736 |
| Linear regression EWAS summary statistics | This manuscript | https://doi.org/10.5281/zenodo.10426429 |
| Bayesian penalised regression EWAS summary statistics | This manuscript | https://doi.org/10.7488/ds/7546 |
| 1000 Genome Phase 3 | Auton et al.[27] | ftp://ftp.1000genomes.ebi.ac.uk/vol1/ftp/data_collections/1000_genomes_project/data |
| **Software and algorithms** | | |
| R | The R Project for Statistical Computing | https://www.r-project.org |
| BayesR+ | Trejo-Banos et al.[11] | https://github.com/ctggroup/BayesRRcmd |
| OSCA | Zhang et al.[13] | https://yanglab.westlake.edu.cn/software/osca/#Overview |
| PLINK | Chang et al.[28] | https://www.cog-genomics.org/plink/ |

### RESOURCE AVAILABILITY

#### Lead contact
Further information and requests for resources and reagents should be directed to and will be fulfilled by the lead contact, Matthew Suderman (matthew.suderman@bristol.ac.uk).

#### Materials availability
This study did not generate new unique reagents.

#### Data and code availability
- According to the terms of consent for GS participants, access to data must be reviewed by the GS Access Committee. Applications should be made to access@generationscotland.org. ALSPAC data access is through a system of managed open access. Submissions and queries should be directed to alspac-data@bristol.ac.uk. For HELIOS, data access request proposals should be directed to helios_science@ntu.edu.sg for the consideration of the HELIOS Study's principal investigators. SABRE data used for this submission will be made available on request to mrclha.swiftinfo@ucl.ac.uk. Further details regarding data sharing can be found on the cohort web pages (https://www.sabrestudy.org/home-2/data-sharing/). Lothian Birth Cohort data access requests can be made by following the information at https://www.ed.ac.uk/lothian-birth-cohorts/data-access-collaboration. Epigenome-wide association statistics from linear models are available via the EWAS Catalog (https://doi.org/10.5281/zenodo.10426429). Epigenome-wide association statistics from Bayesian penalised regression are available at the University of Edinburgh Datashare site (https://doi.org/10.7488/ds/7546). CpGs and weights derived from elastic net regression, Bayesian penalised regression and the combined PCA and elastic net regression strategies are made available at the Edinburgh Datashare site (https://doi.org/10.7488/ds/7546). CpGs and weights for the elastic net regression and 'PCA+elnet'-based approaches from this version of the manuscript are available in Tables S11 and S12.
- All original code has been deposited at Zenodo (https://doi.org/10.5281/zenodo.10154736).
- Any additional information required to reanalyse the data reported in this paper is available from the lead contact upon request.

## EXPERIMENTAL MODEL AND STUDY PARTICIPANT DETAILS

The present study utilised data from human samples only and was not a clinical trial. Sample size estimation was not performed in any cohort, and all available samples from respective recruitment rounds were used in this study. Samples were not allocated to experimental groups, involved in a general procedure across cohorts or subject to general inclusion and exclusion criteria. Study-specific criteria are highlighted in the following sections if applicable. Similarly, consistent medication and health status are not available across all cohorts. To align with the previous literature, CRP levels were log-transformed after adding a constant of 0.01 to prevent undefined values. Figure S9 shows the distribution of CRP levels in the training dataset prior to and following this transformation. Measurements that were outside the median value $\pm$ 4 times the standard deviation were excluded.[6]

### Training cohort: Generationy Scotland

Generation Scotland: Scottish Family Health Study (GS) is a family-structured, population-based cohort study of >24,000 Scottish individuals.[29,30] Recruitment took place between 2006 and 2011. Blood draws were taken during a clinical visit at the study baseline alongside detailed health, lifestyle, cognitive, and sociodemographic data. Whole-blood DNAm was measured using the Illumina Infinium MethylationEPIC array. DNAm was assayed in three distinct sets ($N_{Set1}$ = 5,087, $N_{Set2}$ = 4,450, $N_{Set3}$ = 8,876) and 121 experimental batches. Set 1 contained related individuals. Set 2 consisted of individuals who were unrelated to each other and those in Set 1. Set 3 consisted of related individuals, and individuals related to those in Sets 1 and 2. Set 1 followed a slightly different quality control strategy to Sets 2 and 3, which together followed the same strategy. In Set 1, samples were removed if: (i) $\geq$1% of probes had a detection *p-value* > 0.05 or (ii) there was a disagreement between self-reported sex and methylation-predicted sex. Probes were removed if: (i) $\geq$5% of samples had a bead count <3 or a detection *p-value* > 0.05, (ii) they were non-autosomal or (iii) they overlay any SNPs and/or resided in potential cross-hybridising locations. In Sets 2 and 3, samples were removed if: (i) $\geq$0.5% of probes had a detection *p-value* > 0.01 or (ii) there was a disagreement between self-reported sex and methylation-predicted sex. Probes were excluded if (i) $\geq$5% of samples had a bead count of 3 or less, (ii) $\geq$1% of samples had a detection *p-value* > 0.01, (iii) they were non-autosomal or (iii) they overlay any SNPs and/or resided in potential cross-hybridising locations.

Serum CRP levels (mg/L) were quantified at the University of Glasgow using a commercial high-sensitivity assay on an automated analyser (c311, Roche Diagnostics, UK). Manufacturer's calibration and quality control were employed. There were 17,936 individuals with genome-wide DNAm and CRP measurements following quality control and 752,722 CpG sites were available for analyses.

All components of Generation Scotland received ethical approval from the NHS Tayside Committee on Medical Research Ethics [REC Reference Number: 05/S1401/89]. Generation Scotland has also been granted Research Tissue Bank status by the East of Scotland Research Ethics Service [REC Reference Number: 20-ES-0021], providing generic ethical approval for a wide range of uses within medical research. All participants provided written informed consent.

### Test cohorts

Here, we will present a brief description of each test cohort including the number of individuals with paired CRP and DNAm measurements. Table 1 shows data on demographics and CRP measurements for each test cohort. In keeping with STAR guidelines, we report that sex differences did not impact associations as we observed consistent results between males and females. Of note, different assays were utilised to measure CRP and DNAm across the test cohorts, as indicated in Table 1. ALSPAC, SABRE and LBC all utilised the Illumina 450 K array to assay DNAm and HELIOS used the larger EPIC array.

### Avon longitudinal study of children and parents

Pregnant women who were residing in Avon, UK and with expected dates of delivery from April 1, 1991 to December 31, 1992 were invited to take part in The Avon Longitudinal Study of Children and Parents (ALSPAC).[31–33] The analytical sample in our study included 773 mothers who had DNAm and CRP measured 18 years after the baseline (i.e., after the study pregnancy).[34] Four longitudinal measurements of DNAm and CRP were available for the children at the following ages (in years): age 0, 9, 15 or 17, and 24. DNAm was assayed in peripheral blood at all time-points except for age 0 (cord blood) and age 9 (B-cell-enriched buffy coat). DNAm was also assayed using peripheral blood samples from mothers. Samples across different time-points were distributed in a semi-random manner across slides in order to mitigate batch effects. Data pre-processing was performed using the R package *Meffil*. Samples that had an average probe detection $p \geq 0.01$ were removed, along with those that had sex or genotype mismatches. Probes with detection *p-value*s <0.01 were excluded.

High-sensitivity CRP was quantified using an automated particle-enhanced immunoturbidimetric assay (Roche UK, Welwyn Garden City, UK). CRP was measured in peripheral blood at all time-points apart from cord blood. The number of samples available at each time-point in childhood was 389, 336, 945 and 745, respectively. There were 773 mothers with paired DNAm and CRP measurements following quality control. There were 483,068 CpG sites available for testing.

Study data were collected and managed using REDCap electronic data capture tools hosted at the University of Bristol. REDCap (Research Electronic Data Capture) is a secure, web-based software platform designed to support data capture for research studies.[35] Ethical approval for the study was obtained from the ALSPAC Ethics and Law Committee and the Local Research Ethics

Committees. Informed consent for the use of data collected via questionnaires and clinics was obtained from participants following the recommendations of the ALSPAC Ethics and Law Committee at the time. Consent for biological samples has been collected in accordance with the Human Tissue Act (2004).

### Health for life in Singapore

HELIOS is a population-based cohort of approximately 10,000 Asian men and women living in Singapore. The cohort comprises Singapore citizens or Permanent Residents aged 30–84 years old and excludes pregnant and breastfeeding women, those with major illness requiring hospitalisation or surgery, those who received cancer treatment in the past year or those who participated in drug trials within the month prior to recruitment. Ethnicity was based on self-report data and agreed closely with genetically determined ancestry.[36] The three primary self-reported ethnicities are: (i) Chinese and other East Asian (Chinese), (ii) Malay and other South-East-Asian (Malay) and (iii) South Asian (Indian and other countries from Indian subcontinent). Methylation of genomic DNA was initially quantified in 2,400 samples using the Illumina MethylationEPIC array according to manufacturer's instructions. Bisulfite conversion of genomic DNA was performed using the EZ DNA methylation kit according to manufacturer's instructions (Zymo Research, Orange, CA). Bead intensity was retrieved using the *minfi* software package, and a detection *p-value* of <0.01 was used for marker calling. In total, 846,604 positions were assayed on the array. Markers with call rates beneath 95% were removed (*n* = 8,882 probes). Fifty-eight samples were excluded; two for array scanning failure, 39 for gender inconsistency and 17 duplicates. None of the samples failed the sample call rate criterion (<95%). Marker intensities were normalised by quantile normalisation.[37,38] CRP levels were measured using wide-range CRP technology, which has been proposed as an economic alternative to high-sensitivity CRP screening.[39] CRP was measured from fasting blood samples by the accredited laboratory (QuestLab, Singapore, SAC–SINGLAS ISO 15189:2012) using ADVIA 1800 chemistry system (Siemens Healthcare, Munich, Germany). There were 1,778, 242 and 225 individuals within the Chinese, Malay and Indian ethnicity groups who had paired DNAm and CRP measurements, respectively. The HELIOS dataset contained 837,722 CpG sites.

The HELIOS study was approved by the National Technological University (NTU) Institutional Review Board [IRB-2016-11-030], with written informed consent obtained from each participant before the commencement of the study.

### Southall And Brent REvisited

Southall And Brent REvisited (SABRE) is a population-based study that includes 1,711 first-generation South Asian migrants and 1,762 European-origin individuals, all resident in the UK. Recruitment occurred between 1988 and 1991 in West London, UK. Follow-up clinics were held approximately 20 years later between 2008 and 2013.[40,41] Follow-up samples from male participants were used in the present study. Quality control for DNAm typing was performed using the *meffil.qc.parameters* function in the R package *Meffil*. Baseline and follow-up samples were analyzed together. The following amendments were made to the function's default pipeline: probes and samples with a detection *p-value* <0.1 were excluded, the bead number threshold was reduced from 0.2 to 0.1, sample genotype concordance was reduced from 0.9 to 0.8 and the standard deviation multiple at which sex outliers are identified was raised from 3 to 5 standard deviations.

High sensitivity CRP was measured using an automated platform (c311 Roche Diagnostics, Burgess Hill UK). CRP measurements were available for follow-up samples. There were 588 individuals with paired genome-wide DNAm and CRP available at the follow-up time-point. Of these, 315 and 273 were of European and South Asian ancestry, respectively. The final SABRE DNAm dataset contained 484,781 CpG sites.

The SABRE study was approved by St Mary's Hospital Research Ethics Committee [07/H0712/109] and all participants provided written informed consent.

### The Lothian Birth Cohorts of 1921 and 1936

The Lothian Birth Cohorts of 1921 (LBC1921; *N* = 550) and 1936 (LBC1936; *N* = 1,091) are longitudinal studies of healthy aging. There were five waves of data collection for LBC1921 at mean ages of 79, 83, 87, 90 and 92 years. The LBC1936 has completed six waves of testing at mean ages of 70, 73, 76, 79, 82 and 86 years[42,43] Paired DNAm and CRP data were available at Waves 1–4 for the LBC1936 (age 70–79 years) and Waves 3 and 4 for the LBC1921 (age 87 and 90 years). Low-sensitivity CRP measures were present at Wave 1 of LBC1936 and Waves 3 and 4 of LBC1921. High-sensitivity CRP measurements were available at Waves 2, 3 and 4 of LBC1936.

Methylation was assayed at three separate time-points (Set 1, Set 2 and Set 3). The three sets included 2,195, 996 and 552 samples, respectively. Each set began with 485,512 CpGs. Twenty-three duplicate samples were removed from Set 2. Set 1 and Set 2 had 123 duplicates between them, and a sample was removed from each duplicate pair (108 from Set 1 and 15 from Set 2). Sets 1 and 2 were then combined (as Set1-2). Ten duplicate samples were excluded from Set 3. There were also 31 duplicates between Set 3 and the newly formed Set 1–2. Twenty-six samples were removed from Set 3 and five were removed from Set1-2. The three sets were then combined together (Set1-2-3) and comprised 3,556 samples. Samples and CpGs were filtered on low call rates (CpGs with a detection *p-value* $\geq$ 0.01), with a threshold of 95% for both samples and CpGs. In total, 3,525 samples (spread across Waves and LBC1921/LBC1936) and 470,278 CpGs remained. Finally, sex chromosome probes were removed, leaving a dataset that consisted 459,309 CpGs and 3,525 samples.

Serum CRP was measured from venesected whole-blood samples. CRP levels (mg/L) were quantified using both a high-sensitivity (ELISA; R&D Systems) and a low-sensitivity assay with a dry-slide immuno-rate method on an OrthoFusion 5.1 F.S analyser (Ortho

Clinical Diagnostics). The low-sensitivity assay cannot distinguish between values less than 3 mg/L. Therefore, all readings of <3 mg/L using the low-sensitivity assay were assigned a value of 1.5 mg/L. The number of individuals contributing to our analyses were 885, 756, 536 and 492 at Waves 1, 2, 3 and 4 of the LBC1936, and 170 and 49 at Waves 3 and 4 of the LBC1921, respectively. There were 459,309 CpG sites following quality control.

Ethical approval was obtained from the Multicentre Research Ethics Committee for Scotland (age 70, [MREC/01/0/56]), the Lothian Research Ethics Committee (age 70, [LREC/2003/2/29]), and the Scotland A Research Ethics Committee (ages 73, 76, 79, 82, [07/MRE00/58]). All participants provided written informed consent. Ethical approval was provided by the Lothian Research Ethics Committee for test waves 1–3 at ages 79, 83 and 87 [LREC/1998/4/183, LREC/2003/7/23, 1702/98/4/183] and the Scotland A Research Ethics Committee for test wave 4 at age 90 [10/MRE00/87, 10/MRE00/87]. All participants provided written informed consent.

## METHOD DETAILS

Replication was performed in the sense that multiple diverse cohorts were included, and the performances of DNAm predictors were tested across cohorts in order to infer their generalisabilities. Blinding and randomisation were not applied in this study. Sample size estimation was not used as all available samples at distinct recruitment rounds were utilised. There was no general inclusion or exclusion criteria in this study across the cohorts, excluding a requirement for complete DNAm and CRP data.

### Epigenome-wide association studies on CRP levels
#### Linear regression
Our study analyzed primary data from 17,936 individuals in GS. For comparability, we approximated the modeling approach of a recent EWAS meta-analysis of CRP by Wielscher et al., which was performed in 30 independent studies ($N$ = 22,774).[6] Specifically, 752,722 CpGs were entered separately into linear regression models using the *lm* function in base R. A basic model and fully adjusted model were considered. The basic regression model directly matched the strategy of Wielscher et al.[6] as follows.

$$\log(CRP + 0.01) \sim CpG\ (beta - value) + age + sex + \text{estimated white blood cell proportions} + \text{experimental batch}$$

White blood cell proportions were estimated via the Houseman method.[44] The cell types were B cells, CD4$^+$ T cells, CD8$^+$ T cells, granulocytes, monocytes, and natural killer cells. The proportions of granulocytes were omitted from regression models to avoid collinearity given that the proportions of all six cell types sum to 1. Sensitivity analyses were performed to assess whether the imputation method used to estimate blood cell proportions impacted findings from the basic model. To this end, blood cell proportions were further estimated using the EpiDISH algorithm.[45] The basic model was repeated using blood cell types from this method and compared to that in the main text using the Houseman method. Effect sizes were correlated 88.9% with the main approach (i.e., Houseman method). We also observed a correlation of 90.0% between -$\log_{10}$(*p-value*s). Therefore, the imputation method exhibited a small overall effect on our analyses. Look-up analyses of associations identified in the basic model were performed via the EWAS Catalog.[19]

The fully adjusted model was unique to our study and further considered population structure and five common lifestyle factors given their possible confounding influences on inflammatory and methylation profiles. The lifestyle factors were alcohol consumption, body mass index, deprivation index (Scottish Index of Multiple Deprivation), a methylation-based smoking score (EpiSmokEr),[16] and years of education, and were selected to align with previous publications in the GS cohort.[27,28] Multidimensional scaling (MDS) was applied to GS genotype data to obtain an estimate of population structure. The first 20 MDS components were fitted as fixed-effect covariates. Demographic and covariate data for GS are presented in Table S13. Their associations with CRP are shown in Table S14. The fully adjusted model was as follows.

$$\log(CRP + 0.01) \sim CpG\ (beta - value) + age + sex + \text{estimated white blood cell proportions} + \text{experimental batch}$$
$$+ \text{alcohol consumption} \left(\frac{units}{week}\right) + \log\left(\text{body mass index} \left(\frac{kg}{m^2}\right)\right)$$
$$+ \text{deprivation index (Scottish Index of Multiple Deprivation)} + \text{education (an 11 - category ordinal variable)}$$
$$+ \text{methylation - based smoking score (EpiSmokEr)} + 20\ \text{genetic components (population structure)}$$

### Bayesian penalized regression
Models considering linear associations between outcomes and individual molecular traits do not account for correlation structures within molecular datasets and omitted variable bias.[46] A number of methods have been proposed to overcome these limitations, which include Bayesian penalised regression. The BayesR+ framework implements Bayesian penalised regression and Gaussian mixture-based variance partitioning to inform the molecular architecture of phenotypes.[11] The joint and conditional effects of all 752,722 CpG sites on blood CRP levels were examined. Linear regression models were used to adjust log-transformed CRP levels for age and sex. They were also used to regress CpG beta-values on age, sex, WBC proportions and experimental batch. Residuals from the regressions were scaled to mean zero and unit variance. The prior mixture variances were set to 0.001, 0.01 and 0.1, which corresponds to CpGs that capture 0.1%, 1% and 10% of variation in CRP.[6,7]

The Gibbs algorithm consisted of 10,000 samples (in other words, iterations) and 5,000 samples of burn-in. A thinning of 5 samples was applied in order to reduce autocorrelation. The process left 1,000 samples and was repeated over four chains, which were initialised using a different random number seed for each chain. The final 250 samples (of 1,000) from each chain were combined for downstream analyses, giving a total of 1,000 samples for our analyses. Penalised regression does not consider the identity of probes, which may preclude biological inference. Indeed, the method may select one member from a correlated group of probes as the representative signal in one iteration and another member in the next, thereby distributing their inclusion probabilities. For the epigenome-wide association study, CpGs within 2.5 kilobases and highly correlated (absolute Pearson correlation >0.5) with a lead CpG that had posterior inclusion probability greater than 20% were grouped together. A lead probe was defined as having an initial posterior inclusion probability of greater than 20%. For each probe group, we calculated the proportion of iterations for which at least one probe was included in the model, yielding the group posterior inclusion probability. Included in the model refers to the CpG being assigned to the small, medium or large effect group in a given iteration (i.e., it was not in group 0, corresponding to having no effect on the phenotype). We then calculated the mean (across 1,000 iterations) of the sum of squared regression coefficients for the probe group to give the contribution of the group to the total variance. Finally, we considered groups where their combined posterior inclusion probability was >80% (i.e., at least one member was present in 800 iterations) as significant. We highlighted the lead CpG in each group for clarity. We utilised these parameters and thresholds in line with previous publications on the use of Bayesian penalised regression EWAS for complex traits.[47] It is important to note that the use of different thresholds could lead to small variations in the list of CpGs identified as significant. However, we elected to retain these thresholds in order to maintain consistency with the existing literature.

### Estimating the proportion of variance in CRP levels attributable to genetics and DNAm

BayesR+ was one of two methods implemented in variance component estimation. Here, the same pipeline from EWAS was applied. Variance components estimates were taken as the mean of the sum of squared standardised posterior effect sizes across 1,000 iterations. The 2.5%ile and 97.5%ile (iteration rank 25 and 975) formed the lower and upper bounds of the 95% credible interval, respectively.

Variance partitioning was also performed using OSCA software.[13] The software estimates the phenotype variance captured by a given set of genome-wide molecular trait or omic data (e.g., DNAm) by first constructing an omics relationship matrix (ORM) using all input probes. The ORM describes the covariance pattern across individuals. A univariate linear mixed model is then implemented and fits the ORM as a random effect component. The variance attributed to the molecular component is obtained via restricted maximum likelihood (REML) estimation. Phenotype and methylation data were adjusted as per the BayesR+ strategy. The ORM was constructed from the residuals of all CpG sites ($n$ = 752,722 sites).

Genotyping in Generation Scotland was performed using the Illumina HumanOmniExpressExome-8 Bead Chip. Using Plink,[48] single-nucleotide polymorphisms (SNPs) were excluded on the basis of missing genotype call rate (>2%), departure from the Hardy–Weinberg equilibrium ($p < 1 \times 10^{-6}$) and low minor allele frequency (<1%). Duplicate samples were removed alongside individuals with sex mismatches and missing genotype call rates (>2%). Principal component analysis was performed on GS samples that were combined with 1,092 individuals of the 1000 Genomes (1000G) population.[49] Outliers were defined as those individuals who were more than six standard deviations away from the mean component for the first two principal components. Genomic distance outliers were excluded from all analyses.

For a combined genetic-epigenetic model in BayesR+, additively coded genotypes (i.e., 0, 1 or 2 alleles) at 561,125 SNPs were scaled to mean zero and unit variance as in the strategy described within the main text for methylation. Missing genotypes were mean imputed. The same prior mixture variances were used as in the methylation-based analysis i.e., 0.001, 0.01 and 0.1. Identical phenotype preparations were applied. BayesR+ returned estimates for the proportion of variance in CRP attributed to genome-wide methylation and genetic factors when considered alone and also when conditioned on one another.

In OSCA, a sparse genomic relationship matrix was estimated using genotype data ($n$ = 561,125 markers, –grm-cutoff 0.05). A GRM was initially fitted alone in order to obtain a heritability estimate for CRP. The –multi-orm flag was used in order to fit an ORM (methylation) and a GRM (genetics) as random effect components for restricted maximum likelihood estimation. This allowed for a joint estimation of epigenetic and genetic variance components.

### DNAm prediction of CRP levels

Three primary methods were used to build weighted linear predictors of CRP from genome-wide DNAm: (1) elastic net regression, (2) Bayesian penalised regression and (3) PCA combined with elastic net regression ('PCA+elnet'). The training dataset in all instances was GS ($N$ = 17,936). An adjusted BayesR+ pipeline was applied for prediction when compared to EWAS and variance partitioning. Phenotype and methylation data were adjusted for potential confounders in EWAS and variance component analyses given that the aim was biological inference. Here, log-transformed CRP and DNAm were not adjusted prior to entry in BayesR+; however, they were scaled to mean zero and unit variance. This enabled us to capture reciprocal and external influences on DNAm and CRP, aiding in external prediction efforts. Further, the number of probes in prediction models was restricted to those present in ALSPAC, GS, LBC and SABRE following quality control ($n$ = 374,785 sites). A weighted linear combination of these probes and their effect sizes was used to compute the resultant Bayesian penalised regression-based predictor in all test cohorts. Of note, this probe set ($n$ = 374,785) was used to develop our predictors. However, we also wished to understand how missing CpG sites would impact

prediction reflecting a real-world scenario where external cohorts utilise our weights but may not contain all of the same CpG sites. Whereas, ALSPAC, GS, LBC and SABRE contained all 374,785 sites (by design), the HELIOS dataset possessed 374,592 of the sites thereby lacking 193 sites in the Bayesian predictor.

Elastic net regression is a commonly employed technique to develop DNAm predictors of human traits.[50] Regression models were run using the R package *glmnet*.[51] Log-transformed CRP values were entered as the dependent variable and CpG beta-values (scaled to mean zero and unit variance) served as the independent variables ($n$ = 374,785 sites). The mixing parameter was set to 0.5 (a common default parameter for elastic net models) and 20-fold cross-validation was applied. The model with the lambda value that corresponded to the minimum mean cross-validated error was selected. The optimal model contained 1,468 probes (Table S11). HELIOS harbored 1,466 of these sites while all other test cohorts contained all 1,468 sites. Sensitivity analyses were conducted to assess whether including lifestyle factors as potential features alongside CpG sites impacted model performance. Here, the optimal model selected 589 sites and body mass index alone (and not the other four lifestyle factors included in our study). Wave 2 of the LBC1936 (age 73 years) was retained as the primary test sample. The re-trained elastic net predictor correlated 0.26 with log-transformed CRP levels, which is in contrast with the correlation of ~0.40 in the main analyses. Therefore, incorporating lifestyle factors hampered model performance.

The third method we considered was PCA with elastic net regression. Higgins-Chen et al. enhanced the reliability of epigenetic age estimates by combining PCA with elastic net.[12] In this approach, PCA is applied to all CpG sites of interest in a training dataset in order to identify sets of multi-collinear sites. Elastic net regression is then used to identify an optimal combination of PCs that can predict trait values in external test samples. Here, we first truncated the training DNAm dataset to probes that were deemed significant in a CRP EWAS from Wielscher et al.[6] The authors identified 1,511 independent sites at a *p-value* threshold of $1 \times 10^{-7}$. There were 1,379 of these 1,511 sites in our training dataset (i.e., GS). All test cohorts also contained the same set of 1,379 sites including HELIOS. We further considered an additional series of *p-value* thresholds (beyond $1 \times 10^{-7}$) for probe filtering as sensitivity analyses (see 'Elastic net regression is sufficient to enhance DNAm prediction of CRP'). PCA was performed using the *prcomp* function in R and training data were mean-centred but not scaled as described by Higgins-Chen et al.[12] Elastic net regression was implemented using the parameters described in the previous section (i.e., elastic net regression without PCA) in order to support cross-method comparability. However, it is important to note that the independent variables in this elastic net regression step were PCs rather than individual CpG sites (Table S12). Figure S10 shows associations between the first 20 PCs and relevant covariates in GS.

### Evaluation of DNAm predictors of CRP

The relative performances of five distinct DNAm predictors of CRP were compared. The measures included those derived from elastic net regression (1), Bayesian penalised regression (2) and PCA with elastic net regression (3). The remaining two predictors were derived using EWAS weights from Wielscher et al.[6] (4) and our own linear EWAS (5). Two metrics were used to assess performance. First, Pearson's correlations between log-transformed CRP and DNAm CRP measures were computed across all test cohorts. Second, incremental r-squared ($R^2$) modeling was applied only to a selected test cohort, which was the Lothian Birth Cohort 1936. The cohort was selected in order to align with health outcome association testing. The time-point of Wave 2 (or age 73, $N$ = 756) was selected rather than Wave 1 (age 70, $N$ = 885) as high-sensitivity CRP was available at the former but not latter time-point. The incremental r-squared ($R^2$) was calculated by subtracting the $R^2$ of the full model from that of the null model as shown below.

$$\text{Null model} : \text{Log} - \text{transformed CRP} \sim \text{chronological age} + \text{sex}$$

$$\text{Full model} : \text{Log} - \text{transformed CRP} \sim \text{chronological age} + \text{sex} + \text{DNAm CRP} \, (\text{from one of five methods})$$

### Assessing longitudinal stability of CRP measures

The LBC1936 cohort contained repeat DNAm and CRP measures over four time-points. This cohort had one measure of low-sensitivity CRP at age 70 followed by three high-sensitivity measurements at age 73, 76 and 79 years. The intra-class correlation coefficients of all five DNAm CRP predictors were assessed and compared to that of phenotypic CRP. The *ICC* function in the R package *psych* was used and the 'average random raters' (ICC2k) model was selected to estimate the temporal stabilities of the relevant measures.[52]

### Assessing relationship between assay-measured or DNAm CRP and future CRP measurements

We used the R package *lmerTest* to fit linear mixed-effects models and to regress high-sensitivity CRP measurements on an interaction term between CRP (assay-measured or DNAm) at Wave 2 and chronological age.[53] We co-varied for sex and fitted participant ID as a random effect on the intercept. Here, CRP (assay-measured or DNAm) at Wave 2 was used as the baseline measurement given that Wave 1 contains low-sensitivity measurements alone. Waves 3 and Wave 4 also contain high-sensitivity measurements allowing for a fair comparison of longitudinal relationships across Waves 2, 3 and 4.

### Health outcome association tests

The second Wave of LBC1936 (i.e., at mean age 73 years) was used in health outcome association testing over other cohorts given that the study population was at risk for age-related disease states and frailty phenotypes. The LBC1921 was not used given its smaller analytical sample ($N \leq 170$).

Continuous variables were scaled to mean zero and unit variance. Assay-derived CRP or DNAm CRP served as the independent variable in regression models. Linear regression models were used to examine associations between CRP measures and 21 cardiometabolic and fitness variables (dependent variable). Logistic regression tested for associations between CRP measures and lifetime history of four separate conditions (0 = 'No', 1 = 'Yes'). Cox proportional hazard models assessed the relationship between CRP and all-cause mortality. The time-at-risk ran from age at Wave 2 (~age 73) until the date of recorded death (cases) or the end-of-censor period (controls). All regression models were adjusted for age and sex. Height (in cm) was fitted as an additional fixed-effect covariate for lung function measures and the six meter walk test in linear regression models. A summary of all phenotypes in the LBC1936 is presented in Table S15. Correction for multiple testing was applied using the false discovery rate (FDR $p < 0.05$).[54]

### Polygenic scoring in the LBC1936

A polygenic or genetic score for CRP was computed in LBC1936 participants using PRSice-2 software.[55] LBC1936 DNA samples were genotyped at the Edinburgh Clinical Research Facility using the Illumina 610-Quadv1 array (Wave 1; $n = 1,005$; mean age: $69.6 \pm 0.8$ years; San Diego).[56] SNPs were imputed to the 1000G reference panel (phase 3, version 5).[49] Individuals were excluded on the basis of sex mismatches, family structure, SNP call rates below 95%, and evidence of non-European ancestry. SNPs with a call rate of greater than 98%, minor allele frequency in excess of 1%, and Hardy-Weinberg equilibrium test with $p \geq 0.001$ were retained. An imputation quality score of >0.8 was applied to the imputed set of variants.

Summary statistics from a recent, genome-wide association study on CRP levels were applied to build the genetic score.[20] Only summary data from European samples were applied. An additive weighted genetic score for CRP was constructed from SNPs that passed the genome-wide threshold in the association study ($p < 5 \times 10^{-8}$). Weighted dosages were calculated by multiplying the dose of each risk allele by the effect estimate from the GWAS. The sum of these products produced a genetic score for each individual.

### QUANTIFICATION AND STATISTICAL ANALYSIS

All statistical analysis were performed in R (version 4.0.3).[57] Statistical analyses were at the core of this manuscript and by necessity are detailed in full in Method details. A general overview of the analyses is highlighted here. We denoted the number of samples with 'N' (upper case) throughout the manuscript and other counts, such as the number of CpG sites in analyses, with 'n' (lower case). Measurements were derived from distinct samples across multiple cohorts including Generation Scotland, SABRE and HELIOS. The same sample (i.e., individual) was measured repeatedly across dispersed timepoints in the Lothian Birth Cohorts and ALSPAC. Linear regression models were used to perform epigenome-wide association studies, incremental $R^2$ modeling and association tests between CRP measures and continuous health outcomes. Bayesian penalised regression was also used in epigenome-wide association studies. Logistic regression models tested for associations between CRP measures and self-reported cardiometabolic disease status. Cox proportional hazard models were used to test for associations between CRP measures and all-cause mortality. Pearson's correlation tests assessed the relationship between DNAm CRP and assay-measured CRP. Other statistical methods including feature selection and variance partitioning analyses are detailed in Method details. Due to the very large number of regressions performed, we could not test assumptions for each regression in turn. However, visual inspection of diagnostic plots was performed for key associations and no individual was removed as an influential observation. Convergence of Bayesian models were confirmed by visual inspection of diagnostic plots. Significance thresholds in epigenome-wide association studies were set at $p < 3.6 \times 10^{-8}$, which represents a commonly employed threshold in such analyses.[14] $p$-values in health outcome association tests were adjusted for multiple comparisons using the Benjamini-Hochberg method.[54] The present study utilised all samples with paired DNAm and CRP measurements in the respective cohorts. Table 1 summarises the number of samples utilised in each stage of the analyses. CRP levels are presented using the mean and standard deviation in each cohort or sub-group (Table 1). Figure 2B details mean variance estimates with 95% credible intervals or 95% confidence intervals for Bayesian and frequentist methods, respectively. Pearson's correlation coefficients are shown in Figure 3A and correlation coefficients with 95% confidence intervals are displayed in Figure 3C. Standardised betas, odds ratios and hazard ratios are shown with their 95% confidence intervals in Figure 4, depending on the appropriate statistical test. Center, dispersion and precision measures are detailed in respective figure legends throughout the manuscript.

