## [Document S2. Transparent peer review records for Hillary et al. · Cell Genomics]

Blood-based epigenome-wide analyses of chronic low-grade inflammation
across diverse population cohorts

Author List: Robert F. Hillary¹, Hong Kiat Ng², Daniel L. McCartney¹, Hannah R. Elliott^{3,4}, Rosie M. Walker^{1,5}, Archie Campbell¹, Felicia Huang⁶, Kenan Direk⁷, Paul Welsh⁸, Naveed Sattar⁸, Janie Corley⁹, Caroline Hayward^{1,10}, Andrew M. McIntosh^{1,11}, Cathie Sudlow^{12,13,14}, Kathryn L. Evans¹, Simon R. Cox⁹, John C. Chambers^{2,15}, Marie Loh^{2,15,16,17}, Caroline L. Relton^{3,4}, Riccardo E. Marioni^{1*}, Paul D. Yousefi^{3,4*}, Matthew Suderman^{3,4*,+}

Summary

Initial submission: Received : 11/2/2023

Scientific editor: Judith Nicholson and Laura Zahn

First round of review: Number of reviewers: 2
Revision invited : 1/9/2024
Revision received : 2/9/2024

Second round of review: Number of reviewers: 2
Accepted : 4/3/2024

Data freely available: Yes

Code freely available: Yes

This transparent peer review record is not systematically proofread, type-set, or edited. Special characters, formatting, and equations may fail to render properly. Standard procedural text within the editor's letters has been deleted for the sake of brevity, but all official correspondence specific to the manuscript has been preserved.

Referees' reports, first round of review

Reviewer #1: The authors reported an EWAS for CRP and developed a methylation score for CRP.

General comments:

- * The authors performed comparisons with up-to-date studies in the field using different methodologies.
- * The manuscript is well-written, clear and easy to read, with novelty and conclusions well supported by the results
- * This manuscript aligns with journal aims and scope and recommend to be accepted with minor revisions:

1. BMI and smoking were associated with CRP and changes in DNA methylation (Wielscher et al. 2022), the authors can include them in the main model in EWAS and/or in elastic net models to construct predictors? (i.e. as "forced" unpenalized covariates).
2. For the Bayesian penalized regression in EWAS, why was the posterior inclusion probability 20% used instead of 50% (for instance)? How was this threshold determined?
3. There's no reason/ justification provided for the use of 374,785 CpG sites (in the 5 test cohorts) instead of more than 700,000 CpG sites in the Generation Scotland (GS) to construct the CRP predictors. Was it due to the different platform used for DNAm in these cohorts? Readers may wonder whether the results would be more generalizable if you use all 700,000 CpG sites from the GS.
4. What is the overlap between the 1,468 CpGs from elastic net model and the 1,511 CpGs identified by Wielscher et al.? Are there new CpG sites associated with CRP levels unique in this analysis? I would love to see whether these sites generate any interesting/novel biological interpretation regarding chronic inflammation and methylation, if there's any.

Reviewer #2: In this manuscript, Hillary et al. have conducted a comprehensive study on their newly proposed C-reactive protein (CRP) prediction models using DNA methylation as predictors. The study is highly promising and convincingly executed. The prediction model has been evaluated using several independent testing datasets, demonstrating careful design. I understand the challenges involved in conducting such studies, particularly during revisions. If addressubg any of my suggestions proves difficult, additional clarification in the manuscript would be beneficial. My comments are as follows:

1. Several new blood cell type imputation methods have been proposed. The authors may consider using them as a sensitivity analysis if feasible.
2. The portability of polygenic risk scores is known to vary based on the genetic distance between the training samples and the target population. It would be informative if the manuscript could address whether the derived models are sensitive to ancestry variations.
3. The distribution of log-transformed CRP ($\log[\text{CRP} + 0.01]$) warrants examination to ascertain if it approximates a normal distribution. I understand the authors use this transformation to improve normality, a formal test or visual illustration would be helpful.
4. Clarification is needed regarding the type of platform used in the study. Is this 850K or 450K?

5. Does QQ plot and lambda value look reasonable?

6. When estimating the proportion of variance in CRP levels, it is crucial to know what covariates have been adjusted for. The results might significantly depend on these adjusted covariates. Elaboration on this aspect would be valuable.

6. For the built prediction models, it would be interesting to explore whether incorporating lifestyle factors could significantly improve their performance.

7. For the purpose of reproducibility and promoting open science, the codes should be provided.

8. The resolution of Figure 1 is low (minor).

Authors' response to the first round of review

We are very grateful for the comments provided by the editor and each of the external reviewers of this manuscript. Please see below, in blue, a detailed response to the comments and concerns raised by the reviewers. We hope that the reviewers feel that these have been adequately addressed in the revised manuscript. Line numbers refer to the manuscript file with highlighted changes (in simple markup view).

Reviewer #1:

The authors reported an EWAS for CRP and developed a methylation score for CRP.

General comments:

- * The authors performed comparisons with up-to-date studies in the field using different methodologies.
- * The manuscript is well-written, clear and easy to read, with novelty and conclusions well supported by the results
- * This manuscript aligns with journal aims and scope and recommend to be accepted with minor revisions:

Major comment #1:

"BMI and smoking were associated with CRP and changes in DNA methylation (Wielscher et al. 2022), the authors can include them in the main model in EWAS and/or in elastic net models to construct predictors? (i.e. as "forced" unpenalized covariates)."

Response: We thank the reviewer for their kind and constructive comments on our manuscript. We first highlight that we included body mass index (BMI) and smoking in our fully-adjusted model in the EWAS stage alongside several other pertinent covariates, such as alcohol intake (units/week)¹, socioeconomic deprivation (Scottish index of multiple deprivation) and educational attainment (an 11-category ordinal variable)². Of note, we employed a methylation-based surrogate of smoking behaviour (EpiSmokEr3) rather than self-reported data. Such surrogate biomarkers may provide more accurate measurements than self-report data and outperform self-report variables in health outcome association tests^{4,5}. We observed that lifestyle factors strongly attenuated associations from a basic model that was adjusted for age, sex, blood cell composition and technical variation. Indeed, only 2,805 (or 8.3%) of 33,939 associations from the basic model were also significant in a fully-adjusted model that further accounted for five lifestyle factors and population structure. In other words, 91.7% of associations were lost when co-varying for these factors. As the reviewer outlines, Wielscher et al. demonstrated via mediation analyses that BMI and smoking were strong driving factors of changed CpG methylation in their study on CRP⁶. Therefore, we repeated our analyses to assess the contributions of BMI and smoking, both separately and together, in attenuating associations from the basic model. First, we repeated the basic model with a further adjustment for BMI only. We observed that 72.5% of associations were attenuated to nonsignificance. Effect sizes for associations that were significant in the basic model were attenuated, on average, by 40%. We repeated the same approach for our smoking variable alone and found that 51.5% of associations from the basic model were attenuated to nonsignificance. Effect sizes for associations from the basic model were attenuated, on average, by 20.2%. When BMI and smoking were considered together, 84.9% of associations were attenuated to non-significance. Therefore, BMI and smoking showed strong confounding influences on associations between CpG methylation and CRP, which is in keeping with causal inference analyses in the literature. We report the following additional text on lines 202-209 to highlight our revised analysis:

"Sensitivity analyses showed that body mass index alone attenuated 72.5% of associations from the basic model to non-significance. Effect sizes for associations from the basic model were attenuated, on average, by 40%. Smoking behaviour¹⁶ attenuated 51.5% of associations to non-significance and effect sizes were attenuated, on average, by 20.2%. Co-varying for both body mass index and smoking behaviour (and not other lifestyle factors) attenuated 84.9% of associations from the basic model to non-significance. The substantial attenuation observed after accounting for lifestyle behaviours, and in particular body mass index and smoking, highlights their strong impact on DNAm and chronic low-grade inflammation⁶." We agree that the inclusion of lifestyle factors in weighted linear predictors is an interesting approach. However, one key aim of this manuscript was to construct DNAm-based predictors of CRP and to compare them to existing predictors, which utilise weighted linear combinations of CpG sites alone. Therefore, our main analyses utilised CpG sites as potential features in order to enable fair comparisons and reveal the best-performing methodologies for constructing DNAm surrogates of low-grade inflammation. Similarly, other cohorts would be required to record lifestyle factors with matched phenotype definitions to the training cohort in order to accurately compute the predictor. Nevertheless, we agree that the approach is valuable and we conduct a series of sensitivity analyses to address the reviewer's comment and Comment #7 from Reviewer #2. First, we retrained our elastic net predictor using CpG sites, BMI and smoking behaviour as potential features. BMI was log-transformed and all potential features were scaled to mean zero and unit variance. Generation Scotland remained as our training cohort. We used Wave 2 (age 73) of the Lothian Birth Cohort 1936 as our test sample, given that it was the primary test sample in the main analyses and held high-sensitivity CRP measurements. The optimal model contained 589 probes and BMI. This included 491 probes of the 1,468 that were selected in the optimal model in the main text. When projected into the test sample (LBC1936), the re-trained elastic net predictor correlated 0.26 with logtransformed CRP levels. This is in contrast with a correlation of 0.40 observed with the original elastic net predictor. BMI likely captured and absorbed much of the signal attributed to CpG methylation and CRP. It also appears to be a stronger correlate than smoking behaviour, mirroring findings from our EWAS and existing causal modelling analyses. One potential reason that the inclusion of BMI reduced model performance is that the distribution of BMI and its impact on health is likely to vary between the Generation Scotland (mostly mid-life cohort) and Lothian Birth Cohort 1936 cohorts (older-age cohort). This highlights the strength of using CpG sites alone as features given that the context and measurement of lifestyle factors will vary between cohorts. Second, we retrained our elastic net predictor using CpG sites and all five lifestyle factors included in our fully-adjusted EWAS models. They were alcohol consumption, body mass index, deprivation, education and smoking behaviour. We used the same training and test samples as before. Again, only BMI was selected alongside approximately 600 probes. This predictor also showed a correlation of 0.26. Therefore, considering lifestyle factors alongside CpG sites was not conducive to model performance. We report the following text on lines 823-829 to highlight these additional analyses: "Sensitivity analyses were conducted to assess whether including lifestyle factors as potential features alongside CpG sites impacted model performance. Here, the optimal model selected 589 sites and body mass index alone (and not the other four lifestyle factors included in our study). Wave 2 of the LBC1936 (age 73 years) was retained as the primary test sample. The re-trained elastic net predictor correlated 0.26 with log-transformed CRP levels, which is in contrast with the correlation of ~0.40 in the main analyses. Therefore, incorporating lifestyle factors hampered model performance."

Major comment #2:

"For the Bayesian penalized regression in EWAS, why was the posterior inclusion probability 20% used instead of 50% (for instance)? How was this threshold determined?"

Response: We wish to clarify that we used parameters that were described in previous publications involving Bayesian penalised regression EWAS⁷. We appreciate that, as with any methodology, the use of different thresholds could affect which CpGs are selected for inclusion in the model. Therefore, we highlight these methodological aspects and make them clearer for the reader on lines 772-776:

"We utilised these parameters and thresholds in line with previous publications on the use of Bayesian penalised regression EWAS for complex traits⁴⁶. It is important to note that the use of different thresholds could lead to small variations in the list of CpGs identified as significant. However, we elected to retain these thresholds in order to maintain consistency with the existing literature."

Major comment #3:

"There's no reason/ justification provided for the use of 374,785 CpG sites (in the 5 test

cohorts) instead of more than 700,000 CpG sites in the Generation Scotland (GS) to construct the CRP predictors. Was it due to the different platform used for DNAm in these cohorts? Readers may wonder whether the results would be more generalizable if you use all 700,000 CpG sites from the GS.”

Response: We thank the reviewer for the opportunity to highlight this methodological consideration in our manuscript. As the reviewer identifies, the use of 374,785 CpG sites was due to different platforms being used across the diverse training and test cohorts. Generation Scotland, our training cohort, possessed >700,000 CpG sites. From a biological standpoint, we wished to estimate the marginal association between each available CpG site and CRP levels in the EWAS stage. These CpG sites were assayed using the Illumina EPIC array. However, for prediction, generalisability is of paramount interest. Most existing cohorts, including all test cohorts except HELIOS, utilised the earlier 450 K platform. Therefore, we restricted CpG sites to those present on the EPIC and 450 K platforms to maximise generalisability. Specifically, we restricted CpG sites in our prediction stage to those present in Generation Scotland and the test cohorts ALSPAC, SABRE and the Lothian Birth Cohorts. This restriction also aids in fairer comparisons between training and test cohorts given that many popular prediction algorithms (including elastic net regression and Bayesian penalised regression) model correlation structures between the input probes. A further consideration was that external cohorts might not harbour all 374,785 sites following quality control. To address this, we utilised HELIOS as an additional, external test cohort in order to understand how missingness of CpG sites might alter predictive performances. HELIOS lacked 193 of the 374,785 CpG sites, and we found that it had no detectable effect on the performance of our CRP predictors. We appreciate that these details must be clearer for the reader and we have updated our manuscript to outline these considerations on lines 277-286:

“We note that the training cohort and test cohorts showed variation in terms of the arrays used to measure CRP levels and blood DNAm profiles (Table 1). In the EWAS stage, we utilised all CpGs available in Generation Scotland in order to document as completely as possible relationships between local and global DNAm variation and CRP blood levels. In the prediction stage, we aimed to ensure maximum generalisability to other and older cohorts. Unless otherwise stated, we first restricted the training space to CpG sites that were common to the EPIC and 450 K arrays. Specifically, we restricted CpGs to those present in ALSPAC, GS, SABRE and the LBC cohorts following quality control (n=374,785 sites, see Method details). We also acknowledge that other cohorts may have a slightly different set of CpGs to this list. We held HELIOS as a further external test cohort to understand how missing CpG sites would impact prediction. HELIOS did not contain 193 of these 374,785 sites.”

We have also updated Table 1 and our Methods section to clearly outline differences between DNAm measurement platforms across cohorts. Table 1 is not included here for brevity.

However, we include the following text on lines 658-660 to delineate these differences:

“Of note, different assays were utilised to measure CRP and DNAm across the test cohorts, as indicated in Table 1. ALSPAC, SABRE and LBC all utilised the Illumina 450 K array to assay DNAm and HELIOS used the larger EPIC array.”

Major comment #4:

“What is the overlap between the 1,468 CpGs from elastic net model and the 1,511 CpGs identified by Wielscher et al.? Are there new CpG sites associated with CRP levels unique in this analysis? I would love to see whether these sites generate any interesting/novel biological interpretation regarding chronic inflammation and methylation, if there's any.”

Response: There were 82 CpG sites that overlapped between the elastic net predictor and the CpGs identified by Wielscher et al. This left 1,386 sites (94.4%) that were unique to the elastic net predictor and 1,429 sites (94.6%) that were unique to Wielscher et al. We also examined the overlap in gene annotation. There were 168 genes that were common to both datasets. For context, there were 1,010 unique genes detected by the elastic net model. There were 957 unique annotations in the Wielscher et al. study. This left 842 genes, representing 1,386 sites, that were unique to the elastic net model predictor. Per the reviewer's request, we implemented the gometh() function in the R missMethyl package⁷, which can examine for biological enrichment among CpG sites and can correct for probe density biases. We examined whether there was enrichment for GO (gene ontology) and KEGG (Kyoto Encyclopedia of Genes and Genomes) terms among the CpG sites unique to the elastic net model. No term was detected at an FDR-adjusted p-value < 0.05. Therefore, there was no novel biological interpretation attributable to this prediction model. It likely captures similar signals to the EWAS models, in particular the effect of lifestyle factors on DNAm and chronic inflammation. We report these results here to address the reviewer's comment but do not include it in the manuscript due to space limitations.

Reviewer #2:

In this manuscript, Hillary et al. have conducted a comprehensive study on their newly proposed C-reactive protein (CRP) prediction models using DNA methylation as predictors. The study is highly promising and convincingly executed. The prediction model has been evaluated using several independent testing datasets, demonstrating careful design. I understand the challenges involved in conducting such studies, particularly during revisions. If addressing any of my suggestions proves difficult, additional clarification in the manuscript would be beneficial. My comments are as follows:

Major comment #1:

"Several new blood cell type imputation methods have been proposed. The authors may consider using them as a sensitivity analysis if feasible."

Response: We thank the reviewer for their kind and helpful comments on our manuscript. As the reviewer highlights, there are now alternative methods in the literature to impute blood cell type proportions from methylation data. In our main analyses, we estimated white blood cell proportions via the Houseman method⁸. We adjusted for age, sex, Houseman-estimated white blood cell proportions and technical variation in a basic linear model in our EWAS stage. We detected 33,939 associations at a p-value threshold $< 3.6 \times 10^{-8}$ ⁹. We now extend our approach to estimate white blood cell compositions via the publicly available EpiDISH algorithm¹⁰. We repeated our basic model using estimated blood cell proportions from the EpiDISH algorithm and assessed how well this model agreed with that in the main text, which used the Houseman method. We observed that effect sizes were correlated 88.9% between these models. We also observed a correlation of 90.0% between $-\log_{10}(\text{p-values})$. Lastly, we assessed the proportions of variance in CRP that were captured by estimated cell counts from both methods. The variance explained by Houseman-estimated cell counts was 1.8%. The corresponding estimate was 2.2% for the alternative EpiDISH method¹⁰. Therefore, the imputation method had a small impact on our discovery EWAS. We include the following text on lines 737-743 to highlight these sensitivity analyses:

"Sensitivity analyses were performed to assess whether the imputation method used to estimate blood cell proportions impacted findings from the basic model. To this end, blood cell proportions were further estimated using the EpiDISH algorithm⁴². The basic model was repeated using blood cell types from this method and compared to that in the main text using the Houseman method. Effect sizes were correlated 88.9% with the main approach (i.e. Houseman method). We also observed a correlation of 90.0% between $-\log_{10}(\text{p-values})$. Therefore, the imputation method exhibited a small overall effect on our analyses."

Major comment #2:

"The portability of polygenic risk scores is known to vary based on the genetic distance between the training samples and the target population. It would be informative if the manuscript could address whether the derived models are sensitive to ancestry variations."

Response: We thank the reviewer for highlighting this concern. There have been substantial efforts in genomic research to address whether polygenic scores generalise across diverse ethnic groups. By comparison, research into the portability of DNAm-based predictors is lacking. We are pleased to aid in bridging these knowledge gaps by developing a DNAm-based predictor of CRP, trained in a large homogenous Scottish population, and testing its performance in five diverse cohorts. Our findings show that the performances of CRP predictors are comparable in test cohorts consisting of Scottish individuals (minimal genetic distance to the training sample) and in UK-resident South Asian individuals and individuals of Indian, Malay and Chinese ethnicity in the Singapore-based HELIOS cohort. This represents significant progress in assessing the portability of DNAm-based predictors. Ongoing work will assess the portability of other DNAm-based predictors in these target populations. Future work will include other individuals with large genetic distances from the training sample. To further address the reviewer's comment, we highlight these considerations in our Discussion on lines 454-460:

"Our newly-described predictors performed comparably in residents of the UK and Singapore with diverse self-report ethnicities, which indicates its potential as a biomarker of inflammation in different populations. In contrast to polygenic scores, research into the portability of DNAm-based predictors is limited. There remains a need to expand prediction efforts to individuals across other global regions and of additional ethnic backgrounds and ancestries in order to capture a fuller range of genetic and environmental contexts. It is also essential to include target populations with disparate genetic distances to the training sample in order to reliably document the utility of DNAm-based prediction."

Major comment #3:

"The distribution of log-transformed CRP ($\log[\text{CRP} + 0.01]$) warrants examination to ascertain if it approximates a normal distribution. I understand the authors use this transformation to improve normality, a formal test or visual illustration would be helpful."

Response: We now make this information available as Figure S9 in Document S1. We include the figure below for the reviewer's convenience, which shows CRP levels prior to and following transformation steps and is highlighted for the training cohort, Generation Scotland. We also signpost this information for the reader by including the following text on lines 633-634:

"Figure S9 shows the distribution of CRP levels in the training dataset prior to and following this transformation."

Figure S9. Distributions of CRP levels before and after statistical transformations in the training cohort. CRP levels were trimmed for outliers, which were defined as observations that were outside the median value ± 4 times the standard deviation. CRP levels were also log-transformed to approximate a normal distribution and a constant of 0.01 was added to prevent undefined values. (A) shows values in Generation Scotland prior to these transformation steps. (B) shows values following the transformation. CRP, C-reactive protein.

Major comment #4:

"Clarification is needed regarding the type of platform used in the study. Is this 850K or 450K?"

Response: We apologise that this information was not made clearer in the previous version of the manuscript. We have updated Table 1 to clearly outline the platforms used in the training cohort and each of the test cohorts. For clarity, the EPIC (or 850 K) array was used in Generation Scotland and HELIOS. The earlier 450 K array was used in the remaining test cohorts, which were ALSPAC, SABRE and the LBC cohorts.

Major comment #5:

"Does QQ plot and lambda value look reasonable?"

Response: We now report the Q-Q plots and lambda values for the basic and fully-adjusted linear regression models. The lambda values for the basic and fully-adjusted models were 3.6 and 1.7, respectively. Our large sample size may have allowed us to detect a very large number of associations with significant but small effects. However, the inflation in these models is likely explained by the strong confounding influences of lifestyle behaviours on DNAm and CRP levels, as well as unknown confounding influences. A Bayesian penalised regression model was also included as it can control genomic inflation in EWAS11. As well as reporting the lambda values in the main text, we include the following text on lines 201-209 in our revised manuscript to highlight the inflation seen in our linear regression models: "Q-Q plots for the basic and fully-adjusted models are shown in Figure S1. Sensitivity analyses showed that body mass index alone attenuated 72.5% of associations from the basic model to non-significance. Effect sizes for associations from the basic model were attenuated, on average, by 40%. Smoking behaviour¹⁶ attenuated 51.5% of associations to nonsignificance

and effect sizes were attenuated, on average, by 20.2%. Co-varying for both body mass index and smoking behaviour (and not other lifestyle factors) attenuated 84.9% of associations from the basic model to non-significance. The substantial attenuation observed after accounting for lifestyle behaviours, and in particular body mass index and smoking, highlights their strong impact on DNAm and chronic low-grade inflammation⁶. We include the Q-Q plots as a new Figure S1 and update our documents accordingly:

Figure S1. Q-Q plots for the basic and fully-adjusted models in the linear regression EWAS. EWAS, epigenome-wide association study.

Major comment #6:

“When estimating the proportion of variance in CRP levels, it is crucial to know what covariates have been adjusted for. The results might significantly depend on these adjusted covariates. Elaboration on this aspect would be valuable.”

Response: We thank the reviewer for the opportunity to clarify this portion of our variance partitioning analyses. We agree that it is important to make clearer the covariates used when estimating the proportion of variance in CRP levels. We include the following text on lines 226-229 to clarify the covariate strategy employed in our analyses:

“Variance estimates can be affected by the covariates used in the associated models. Logtransformed CRP levels were adjusted for age and sex prior to entry in variance partitioning analyses. CpG beta-values were also adjusted for age, sex, WBC proportions and experimental batch.”

Major comment #7:

“For the built prediction models, it would be interesting to explore whether incorporating lifestyle factors could significantly improve their performance.”

Response: We agree that there is inherent value in incorporating lifestyle factors into weighted linear predictors. However, our primary goal was to construct DNAm-based predictors for CRP and compare them with existing CpG site-based predictors. In light of our original aims, we retain CpG site-based predictors in our main elastic net approach. However, we conducted sensitivity analyses to address the reviewer’s comment and Comment #1 from Reviewer #1. First, we retrained the elastic net predictor using CpG sites, BMI, and smoking behaviour as potential features, given that these lifestyle factors show the strongest mediating effect between CpG methylation and CRP levels⁶. Generation Scotland served as our training cohort and our test cohort was Wave 2 of the Lothian Birth Cohort 1936 (age 73 years), given that it was the primary test sample in the main text. The optimal model selected 589 probes and BMI. The retrained predictor showed a correlation of 0.26 with CRP levels, contrasting the correlation of 0.40 observed with our original predictor (based on CpG sites alone). In a second analysis, we retrained the elastic net predictor using CpG sites and all five lifestyle factors included in our study (alcohol consumption, BMI, deprivation, education, and smoking behaviour). Only BMI was selected alongside approximately 600 probes. This predictor showed a correlation of 0.26 with log-transformed CRP levels when projected into the test cohort. BMI is a strong confounder in the context of DNAm and CRP, and appeared to capture much of the signal attributed to CpG methylation and CRP variability. One potential reason why the inclusion of BMI hampered model performance is that the distribution of BMI and its

impact on health will vary between the training and target populations. This highlights an inherent weakness in using lifestyle factors as additional features in such predictors and emphasises the strength of using DNAm data alone. Additional analyses are reported on lines 823-829 to emphasize these findings:

"Sensitivity analyses were conducted to assess whether including lifestyle factors as potential features alongside CpG sites impacted model performance. Here, the optimal model selected 589 sites and body mass index alone (and not the other four lifestyle factors included in our study). Wave 2 of the LBC1936 (age 73 years) was retained as the primary test sample. The re-trained elastic net predictor correlated 0.26 with log-transformed CRP levels, which is in contrast with the correlation of ~ 0.40 in the main analyses. Therefore, incorporating lifestyle factors hampered model performance."

Major comment #8:

"For the purpose of reproducibility and promoting open science, the codes should be provided."

Response: We are pleased to make all code available via Zenodo (DOI: <https://doi.org/10.5281/zenodo.10154736>). We also make our EWAS summary statistics and predictors publicly available and include links to all resources in the Key resources table within STAR ★ Methods.

Minor comment #1:

"The resolution of Figure 1 is low (minor)."

Response: We thank the reviewer for highlighting this. We have revised our Figures to ensure that their resolutions meet the requirements of the journal.

References

1. Bell, S., Mehta, G., Moore, K. & Britton, A. Ten-year alcohol consumption typologies and trajectories of C-reactive protein, interleukin-6 and interleukin-1 receptor antagonist over the following 12 years: a prospective cohort study. *Journal of Internal Medicine* 281, 75-85 (2017).
2. Kershaw, K.N., Mezuk, B., Abdou, C.M., Rafferty, J.A. & Jackson, J.S. Socioeconomic position, health behaviors, and C-reactive protein: a moderated-mediation analysis. *Health Psychol* 29, 307-16 (2010).
3. Bollepalli, S., Korhonen, T., Kaprio, J., Anders, S. & Ollikainen, M. EpiSmokEr: a robust classifier to determine smoking status from DNA methylation data. *Epigenomics* 11, 1469-1486 (2019).
4. Zhang, Y. et al. Smoking-associated DNA methylation markers predict lung cancer incidence. *Clinical epigenetics* 8, 1-12 (2016).
5. Corley, J. et al. Epigenetic signatures of smoking associate with cognitive function, brain structure, and mental and physical health outcomes in the Lothian Birth Cohort 1936. *Transl Psychiatry* 9, 248 (2019).
6. Wielscher, M. et al. DNA methylation signature of chronic low-grade inflammation and its role in cardio-respiratory diseases. *Nature Communications* 13, 2408 (2022).
7. McCartney, D.L. et al. Blood-based epigenome-wide analyses of cognitive abilities. *Genome Biology* 23, 26 (2022).
8. Houseman, E.A. et al. DNA methylation arrays as surrogate measures of cell mixture distribution. *BMC Bioinformatics* 13, 86 (2012).
9. Saffari, A. et al. Estimation of a significance threshold for epigenome-wide association studies. *Genet Epidemiol* 42, 20-33 (2018).
10. Teschendorff, A.E., Breeze, C.E., Zheng, S.C. & Beck, S. A comparison of reference-based algorithms for correcting cell-type heterogeneity in Epigenome-Wide Association Studies. *BMC Bioinformatics* 18, 105 (2017).
11. Trejo Banos, D. et al. Bayesian reassessment of the epigenetic architecture of complex traits. *Nat Commun* 11, 2865 (2020).

Referees' reports, second round of review

Reviewer #1: Thank you for your careful response to review, and congratulations on beautiful work!

Reviewer #2: Thank you for addressing most of my comments. I really appreciate your

explanation regarding the transferability of prediction models based on DNA methylation. I have one follow-up minor question. I am curious whether the QQ plot looks better if you adjust for the estimated blood cell proportions from the EpiDISH.

Authors' response to the second round of review

We are very grateful for the comments provided by the editor and each of the external reviewers of this manuscript. Please see below, in blue, a detailed response to the remaining comment raised by reviewer #2. We hope that the reviewer and editor feel that remaining concerns have been adequately addressed in the revised manuscript.

Reviewer #2:

Major comment #1:

"Thank you for addressing most of my comments. I really appreciate your explanation regarding the transferability of prediction models based on DNA methylation. I have one followup minor question. I am curious whether the QQ plot looks better if you adjust for the estimated blood cell proportions from the EpiDISH."

Response: We thank the reviewer for their kind and constructive comments on our manuscript. We repeated our basic and fully-adjusted linear EWAS models using EpiDISH to estimate cell type proportions instead of the Houseman method. All other covariates remained the same. We re-calculated lambda values and regenerated Q-Q plots using our EpiDISH-informed analyses. The lambda values were largely unchanged. The lambda value for the basic model went from 3.6 (using the Houseman algorithm) to 3.3 (using the EpiDISH algorithm). The lambda value for the fully-adjusted model went from 1.7 to 1.6, respectively. Q-Q plots are shown below for the EpiDISH algorithm (Figure 1), and mirror those of the main analytical strategy (Figure 2). Given these lines of evidence, and those in our previous rebuttal, we do not believe that the selected method for estimating cell type profiles made a considerable impact on our analyses.

Figure 1. Q-Q plots for the basic and fully-adjusted models in the linear regression EWAS when using EpiDISH-estimated cell type proportions. EWAS, epigenome-wide association study.

Figure 2. Q-Q plots for the basic and fully-adjusted models in the linear regression EWAS when using Houseman-estimated cell type proportions. EWAS, epigenome-wide association study.